# Smoothed Online Classification can be Harder than Batch Classification

**Vinod Raman** *
Department of Statistics
University of Michigan
Ann Arbor, MI 48104
vkraman@umich.edu

**Unique Subedi** *
Department of Statistics
University of Michigan
Ann Arbor, MI 48104
subedi@umich.edu

**Ambuj Tewari**
Department of Statistics
University of Michigan
Ann Arbor, MI 48104
tewaria@umich.edu

## Abstract

We study online classification under smoothed adversaries. In this setting, at each time point, the adversary draws an example from a distribution that has a bounded density with respect to a fixed base measure, which is known apriori to the learner. For binary classification and scalar-valued regression, previous works [Haghtalab et al., 2020, Block et al., 2022] have shown that smoothed online learning is as easy as learning in the iid batch setting under PAC model. However, we show that smoothed online classification can be harder than the iid batch classification when the label space is unbounded. In particular, we construct a hypothesis class that is learnable in the iid batch setting under the PAC model but is not learnable under the smoothed online model. Finally, we identify a condition that ensures that the PAC learnability of a hypothesis class is sufficient for its smoothed online learnability.

## 1 Introduction

Classification is a canonical machine learning task where the goal is to classify examples in $\mathcal{X}$ into one of the possible classes in $\mathcal{Y}$. There are two common classification settings based on how the data is available to the learner: batch and online. In the batch setting, the learner is provided with a fixed set of training samples that are used to train a classifier, which is then deployed to make predictions on new, real-world examples [Vapnik and Chervonenkis, 1971, Natarajan, 1989]. On the other hand, data arrives sequentially in the online setting and predictions need to be made in each round [Littlestone, 1987, Daniely et al., 2011]. The batch setting is often studied under the iid assumption, whereas the stream can be fully adversarial in the online setting.

For binary classification (i.e. $|\mathcal{Y}| = 2$), the batch learnability of a hypothesis class $\mathcal{H} \subseteq \mathcal{Y}^{\mathcal{X}}$ under the PAC model [Valiant, 1984] is characterized in terms of a combinatorial parameter called the Vapnik-Chervonenkis (VC) dimension of $\mathcal{H}$. On the other hand, the Littlestone dimension characterizes the learnability of $\mathcal{H}$ under the adversarial online model [Littlestone, 1987, Ben-David et al., 2009]. As Haghtalab et al. [2020] remark, the latter characterization is often interpreted as an impossibility result because even simple classes like 1-dimensional thresholds have infinite Littlestone dimension. This hardness result arises mainly because the adversary can deterministically choose hard examples, even possibly adapting to the learner's strategy. One way to circumvent this hardness result is to consider a smoothed online model, where the adversary has to choose and draw examples from sufficiently anti-concentrated distributions [Rakhlin et al., 2011, Haghtalab, 2018, Haghtalab et al., 2020, Block et al., 2022]. This idea is inspired from the seminal work by Spielman and Teng [2004], who showed that the smoothed analysis of the simplex method yields a polynomial time complexity in the input size, instead of the known worst-case exponential time complexity.

---

*Equal Contribution

38th Conference on Neural Information Processing Systems (NeurIPS 2024).

In smoothed online classification, a learner plays a game with the adversary over $T \in \mathbb{N}$ rounds. Before the game begins, the adversary reveals a base measure $\mu$ over $\mathcal{X}$ and an anti-concentration parameter $\sigma > 0$ to the learner. The distribution $\mu$ can be fairly non-informative such as uniform when applicable for $\mathcal{X}$. Then, in each round $t \in [T]$, the adversary picks a labeled sample $(x_t, y_t) \in \mathcal{X} \times \mathcal{Y}$, where $x_t$ is drawn from a $\sigma$-smooth distribution $\nu_t$ with respect to $\mu$. That is, $\nu_t(E) \leq \mu(E)/\sigma$ for all measurable subsets $E$ in $\mathcal{X}$. The adversary then reveals $x_t$ to the learner, who makes a prediction $\hat{y}_t \in \mathcal{Y}$. Finally, the adversary reveals the true label $y_t \in \mathcal{Y}$ and the learner suffers the loss $\mathbb{1}\{\hat{y}_t \neq y_t\}$. Given a hypothesis class $\mathcal{H} \subseteq \mathcal{Y}^{\mathcal{X}}$, the goal of the learner is to minimize the regret, the difference between its cumulative loss and the best possible cumulative loss over hypotheses in $\mathcal{H}$.

The smoothed online model interpolates between the iid setting ($\sigma = 1$) and the adversarial setting ($\sigma = 0$). When $|\mathcal{Y}| = 2$, Haghtalab [2018] and Haghtalab et al. [2020] showed that all VC classes are learnable in the smoothed setting with the regret $O(\sqrt{T \, \mathrm{VC}(\mathcal{H}) \log(T/\sigma)})$. Extending this result to real-valued regression, where $\mathcal{Y} = [-1, 1]$, with the absolute-value loss, Block et al. [2022] showed that the finiteness of the fat-shattering dimension [Bartlett et al., 1996, Alon et al., 1997] of $\mathcal{H} \subseteq \mathcal{Y}^{\mathcal{X}}$ is a sufficient condition for smoothed online learnability. Since the finiteness of VC and fat-shattering dimensions are characterizations of learnability under the PAC model, these results suggest that smoothed online learning may be as easy as batch learning.

In this work, we study smoothed online classification for arbitrary label spaces $\mathcal{Y}$ under *oblivious* adversary – one that picks $\sigma$-smooth distirbutions $\nu_1, .., \nu_T$, samples $x_1 \sim \nu_1, ..., x_T \sim \nu_T$ independently, and finally picks the labels $y_1, ..., y_T$, all before the game begins. This obvlious model has been studied in the past Wu et al. [2023], but is slightly different than the one studied by [Haghtalab, 2018, Haghtalab et al., 2020, Block et al., 2022] (see Section 2.1 for more details). For this model, we show that smoothed online classification continues to be as easy as batch classification when $|\mathcal{Y}| < \infty$. However, when $|\mathcal{Y}|$ is not finite, we show that smoothed online classification can be *harder* than batch classification. We note that there has been recent interest in studying multiclass learnability when $|\mathcal{Y}|$ is unbounded [Brukhim et al., 2022, Hanneke et al., 2023, Pabbaraju, 2024]. Studying infinite label spaces is important for understanding when one can establish learning guarantees independent of the label size. This is quite a practical question as many modern machine learning paradigms have massive label space, such as in face recognition and protein structure prediction, where the dependence of label size in learning bounds would be undesirable.

**Theorem** (Informal) Let $\mathcal{X} = [0, 1]$. Then, there exists $\mathcal{H} \subseteq \mathcal{Y}^{\mathcal{X}}$ that is PAC learnable but not learnable in the smoothed online setting with $\mu = \mathrm{Uniform}(\mathcal{X})$ and $\sigma = 1$.

We also provide a quantitative version of this theorem that shows a regret lowerbound linear in $T$ even when $|\mathcal{Y}| < \infty$ but bigger than $2^{T \log(T)}$. Note that this is tight up to a factor of $\log T$ because we prove a sublinear upperbound as long as $|\mathcal{Y}| = 2^{o(T)}$ in Section 4. To prove these results, we exploit the large size of the label space to construct a hypothesis class $\mathcal{H} \subseteq \mathcal{Y}^{\mathcal{X}}$ such that for every $h \in \mathcal{H}$, its output on a finite subset of $\mathcal{X}$ effectively reveals its identity. We then show that such a hypothesis class has a sample compression scheme of size 1. Then, the result "compression implies learning" by David et al. [2016] shows that $\mathcal{H}$ is PAC learnable. However, we show that even the adversary that generates iid samples from $\mathrm{Uniform}(\mathcal{X})$ can construct a difficult stream for the learner. Our construction is inspired by the hypothesis class from [Hanneke et al., 2024, Claim 5.4]. However, a key challenge in our construction is the fact that the adversary does not have full control over the sequence of examples the learner will observe (due to $\sigma$-smoothness) whereas the adversary in Hanneke et al. [2024] can pick hard examples deterministically.

In light of this hardness result, we identify a sufficiency condition for smoothed online learnability. To do so, let $\mathrm{B}(\mu, \sigma)$ denote the set of all $\sigma$-smooth distributions with respect to $\mu$. For any $x_1, \ldots, x_n \in \mathcal{X}$, let us define an empirical metric on $d_n$ on $\mathcal{H}$ as $d_n(h_1, h_2) = n^{-1} \sum_{i=1}^n \mathbb{1}\{h_1(x_i) \neq h_2(x_i)\}$. Define $\mathcal{N}(\varepsilon, \mathcal{H}, d_n)$ to be the covering number of $\mathcal{H}$ under metric $d_n$. Then, $\mathcal{H}$ is said to have uniformly bounded expected empirical metric entropy (UBEME) if $\sup_{n \in \mathbb{N}} \sup_{\nu_{1:n} \in \mathrm{B}(\mu, \sigma)} \mathbb{E}_{x_{1:n} \sim \nu_{1:n}} [\mathcal{N}(\varepsilon, \mathcal{H}, d_n)] < \infty$ for every fixed $\varepsilon > 0$. We show that if $\mathcal{H}$ has the UBEME property, then it online learnable under smoothed adversaries.

**Theorem** (Informal) If $\sup_{n \in \mathbb{N}} \sup_{\nu_{1:n} \in \mathrm{B}(\mu, \sigma)} \mathbb{E}_{x_{1:n} \sim \nu_{1:n}} [\mathcal{N}(\varepsilon, \mathcal{H}, d_n)] < \infty$ for every fixed $\varepsilon, \sigma > 0$, then $\mathcal{H}$ is smoothed online learnable under the base measure $\mu$.

When $|\mathcal{Y}| = 2$, Haussler's packing lemma implies that $\mathcal{N}(\varepsilon, \mathcal{H}, d_n) \leq \left(41\,\varepsilon^{-1}\right)^{\mathrm{VC}(\mathcal{H})}$ [Haussler, 1995]. That is, every class with finite VC satisfies UBEME, and thus our sufficiency condition recovers the result from Haghtalab [2018] on the smoothed learnability of VC classes. For $|\mathcal{Y}| < \infty$, we generalize the packing lemma to show that $\mathcal{N}(\varepsilon, \mathcal{H}, d_n) \leq (\frac{22\,|\mathcal{Y}|}{\varepsilon})^{\mathrm{G}(\mathcal{H})}$, where $\mathrm{G}(\mathcal{H})$ is the graph dimension of $\mathcal{H}$. This inequality shows that PAC learnability of $\mathcal{H}$ is sufficient for its smoothed online learnability when $|\mathcal{Y}| < \infty$.

A key contribution of our sufficiency result is going beyond VC and Graph dimension and giving the weaker sufficiency condition than the worst-case empirical entropy. Indeed, in Section 4, we show that our sufficiency condition still provides meaningful upperbounds even when the VC and Graph dimensions are infinite. To prove our sufficiency result, we show that UBEME implies the bounded metric entropy of $\mathcal{H}$ with respect to the base measure $\mu$. That is, for $d_\mu(h_1, h_2) = \mathbb{P}_{x \sim \mu}[h_1(x) \neq h_2(x)]$, we have $\mathcal{N}(\varepsilon, \mathcal{H}, d_\mu) < \infty$ for every fixed $\varepsilon > 0$. Then, we use algorithmic ideas from Haghtalab [2018] that involve running multiplicative weights using the cover of $\mathcal{H}$ under $d_\mu$. Unfortunately, when $|\mathcal{Y}|$ is unbounded, the sufficiency condition (the UBEME of $\mathcal{H}$) is not necessary. This is demonstrated by constant functions $\mathcal{H} = \{x \mapsto a : a \in \mathbb{N}\}$ that is easy to learn but $\mathcal{N}(\varepsilon, \mathcal{H}, d_1) = \infty$.

Given our separation and sufficiency results, it is natural to ask for a characterization of learnability for smoothed online classification. Any meaningful characterization must be a joint property of both $\mathcal{H}$ and $\mu$. This is because choosing $\mu$ to be a Dirac distribution will make any $\mathcal{H}$, even the set of all measurable functions from $\mathcal{X}$ to $\mathcal{Y}$, trivially learnable. Since the most natural joint complexity measure of $\mathcal{H}$ and $\mu$ is $\mathcal{N}(\varepsilon, \mathcal{H}, d_\mu)$, one might ask whether $\mathcal{N}(\varepsilon, \mathcal{H}, d_\mu) < \infty$ for every $\varepsilon > 0$ is necessary and sufficient for smoothed online learnability of $\mathcal{H}$ under $\mu$. Surprisingly, we show that this condition is neither necessary nor sufficient (see Theorem 4.3). These results highlight the difficulty of characterizing learnability in the smoothed setting.

## 2    Preliminaries

Let $\mathcal{X}$ denote the instance space and $\mathcal{Y}$ denote the label space. We assume that $(\mathcal{X}, \Sigma)$ is a measurable space and let $\Pi(\mathcal{X})$ denote the set of all probability measures on $(\mathcal{X}, \Sigma)$. Let $\mathcal{H} \subseteq \mathcal{Y}^{\mathcal{X}}$ denote an arbitrary hypothesis class consisting of predictors $h : \mathcal{X} \to \mathcal{Y}$. For any $T \in \mathbb{N}$, we use the notation $z_{1:T}$ to denote the sequence $\{z_t\}_{t=1}^{T}$. Finally, we let $[N] := \{1, 2, \ldots, N\}$.

### 2.1    Smoothed Online Learning

In the smoothed online model, an adversary plays a sequential game with the learner over $T$ rounds. Before the game begins, the adversary reveals a base measure $\mu \in \Pi(\mathcal{X})$ and a scalar $\sigma > 0$ to the learner. As mentioned before, $\mu$ can be non-informative measures such as uniform if $\mathcal{X}$ is totally-bounded. Then, in each round $t \in [T]$, an adversary picks a labeled sample $(x_t, y_t) \in \mathcal{X} \times \mathcal{Y}$, where $x_t$ is drawn from a distribution $\nu_t \in \Pi(\mathcal{X})$ that satisfies $\nu_t(E) \leq \frac{\mu(E)}{\sigma}$ for every $E \in \Sigma$. The adversary then reveals $x_t$ to the learner $\mathcal{A}$. Using all the past examples $(x_1, y_1), \ldots, (x_{t-1}, y_{t-1})$, the learner then makes a potentially randomized prediction $\mathcal{A}(x_t)$. The adversary then reveals the true label $y_t \in \mathcal{Y}$ and the learner suffers the loss $\mathbb{1}\{\mathcal{A}(x_t) \neq y_t\}$. Given a hypothesis class $\mathcal{H} \subseteq \mathcal{Y}^{\mathcal{X}}$, the goal of the learner is to output predictions $\mathcal{A}(x_t)$ that minimizes the regret, which is the difference between its cumulative loss and the best possible cumulative loss over hypotheses in $\mathcal{H}$. To formally define the regret, let $\mathrm{B}(\mu, \sigma) := \{\nu \in \Pi(\mathcal{X}) : \nu(E) \leq \mu(E)/\sigma \quad \forall E \in \Sigma\}$ denote the set of all $\sigma$-smooth distributions on $\mathcal{X}$ with respect to $\mu$. Given $\mathcal{H} \subseteq \mathcal{Y}^{\mathcal{X}}$, the worst-case expected regret of an algorithm $\mathcal{A}$ is defined as

$$\mathrm{R}_{\mathcal{A}}^{\mu, \sigma}(T, \mathcal{H}) := \sup_{\nu_1, \ldots, \nu_T \in \mathrm{B}(\mu, \sigma)} \mathbb{E}_{x_{1:T} \sim \nu_{1:T}} \left[ \sup_{y_{1:T}} \left( \sum_{t=1}^{T} \mathbb{E}_{\mathcal{A}} \left[ \mathbb{1}\{\mathcal{A}(x_t) \neq y_t\} \right] - \inf_{h \in \mathcal{H}} \sum_{t=1}^{T} \mathbb{1}\{h(x_t) \neq y_t\} \right) \right].$$

Note that as $\sigma \to 0$, the set $\mathrm{B}(\mu, \sigma)$ contains all Dirac distributions on $\mathcal{X}$. This amounts to replacing $\sup_{\nu_1, \ldots, \nu_T \in \mathrm{B}(\mu, \sigma)} \mathbb{E}_{x_{1:T} \sim \nu_{1:T}}[\cdot]$ operator in the definition of regret above by $\sup_{x_{1:T}}$, which yields the expected regret of $\mathcal{A}$ in the fully adversarial model under an oblivious adversary. Thus, our adversary is a generalization of the online oblivious adversary for the smoothed setting. Given this definition of expected regret, we adopt the minimax perspective to define the learnability of a hypothesis class.

**Definition 1** (Smoothed Online Learnability). *The class $\mathcal{H} \subseteq \mathcal{Y}^{\mathcal{X}}$ is learnable in the smoothed online setting if and only if for every $\sigma > 0$ and $\mu \in \Pi(\mathcal{X})$, we have*

$$\inf_{\mathcal{A}} \mathrm{R}_{\mathcal{A}}^{\mu,\sigma}(T, \mathcal{H}) = o(T).$$

Our worst-case expected regret is defined with respect to an *oblivious* adversary that picks the entire stream $(x_1, y_1), \ldots, (x_T, y_T)$ before the game begins. Moreover, the sequence of distributions $\nu_1, \ldots, \nu_T$ has to be chosen upfront before the sampling step $(x_1, \ldots, x_T) \sim (\nu_1, \ldots, \nu_T)$. That is, the distribution $\nu_t$ cannot depend on the realization of previous instances $x_1, \ldots, x_{t-1}$. This ensures that the instances $x_1, \ldots, x_T$ are independent random variables. One can also consider an oblivious adversary, where $\nu_t$ can depend on the past instances $(x_1, \ldots, x_{t-1})$ sampled from $(\nu_1, \ldots, \nu_{t-1})$. Since the primary contribution of this work is the hardness result in Section 3, we only focus on the case where $\nu_1, \ldots, \nu_T$ are chosen upfront. As the first adversary is a special case of the second adversary, our hardness result also applies for the second adversary. The fully general setting of adaptive adversaries where $\nu_t$ can depend on the *entire* history of the game up to time point $t - 1$ has been studied extensively in [Haghtalab et al., 2020, Block et al., 2022], where the dependence among $x_1, \ldots, x_T$ is handled through coupling.

There are three natural choices for how $y_t$ may be selected by an oblivious adversary. In the first choice, $y_t$ may depend only on the $x_t \sim \nu_t$. In the second choice, $y_t$ may depend on prefix on $x_1 \sim \nu_1, \ldots, x_t \sim \nu_t$. Finally, in the last choice, $y_t$ may depend on the entire sample $x_1 \sim \nu_1, \ldots, x_T \sim \nu_T$. The first choice is considered in Haghtalab [2018]and the second by Haghtalab et al. [2020], Block et al. [2022]. In this work, we focus on the third choice, which has been considered by Wu et al. [2023]. This choice is natural because $\sigma$-smoothness is just a property of the instances and should not impact how the labels are selected. Since the third choice is the strongest, our sufficiency result in Section 4 holds for the first two choices. However, establishing the separation result for the first two choices remains an open question.

## 2.2 PAC Learning and Sample Compression Schemes

In contrast to existing work, we establish a separation between smoothed online learnability and batch learnability. The notion of batch learnability we consider is PAC learnability, a canonical model in statistical learning theory. See Appendix A for a complete definition. To prove the agnostic PAC learnability of hypothesis classes, we use the relationship between learnability and the existence of sample compression schemes.

A compression scheme $(\kappa, \rho)$ consists of a compression function $\kappa$ and a reconstruction function $\rho$. The compression function $\kappa : \cup_{i \geq 1} (\mathcal{X} \times \mathcal{Y})^i \to \cup_{i \geq 1} (\mathcal{X} \times \mathcal{Y})^i$ maps a sample $S = \{(x_1, y_1), \ldots, (x_n, y_n)\}$ to a subsample $S' \subseteq S$. The reconstruction function $\rho : \cup_{i \geq 1} (\mathcal{X} \times \mathcal{Y})^i \to \mathcal{Y}^{\mathcal{X}}$ takes $S'$ as input and outputs a function $f \in \mathcal{Y}^{\mathcal{X}}$. We define the size of the compression scheme $(\kappa, \rho)$ on a sample $S$ to be $|S'|$, where $\kappa(S) = S'$. We let the quantity $k(n)$ denote the maximum size of the compression scheme on all samples $S$ such that $|S| = n$. A hypothesis class $\mathcal{H} \subseteq \mathcal{Y}^{\mathcal{X}}$ has a compression scheme $(\kappa, \rho)$ of size $k(n)$ if for every sample $S = \{(x_1, h(x_1)), \ldots, (x_n, h(x_n))\}$ for some $h \in \mathcal{H}$, we have $f = \rho(\kappa(S))$ such that $f(x_i) = h(x_i)$ for all $i \in [n]$.

The compression scheme in David et al. [2016] is slightly more general as their compression function $\kappa$ can output $(S', b)$ where $b$ is a finite bitstring. However, the restricted notion of a compression scheme without $b$ is sufficient for our purpose. The following Theorem shows that the existence of sample compression schemes for $\mathcal{H}$ implies agnostic PAC learnability of $\mathcal{H}$.

**Theorem 2.1** (Compression $\implies$ Learnability [David et al., 2016]). *Let $(\kappa, \rho)$ be a sample compression scheme for $\mathcal{H} \subseteq \mathcal{Y}^{\mathcal{X}}$ of size $k(n)$ and define $f_S = \rho(\kappa(S))$ for any $S \in (\mathcal{X} \times \mathcal{Y})^n$. Then, for every $\mathcal{D}$ on $\mathcal{X} \times \mathcal{Y}$ and $n \in \mathbb{N}$ such that $k(n) \leq n/2$, with probability at least $1 - \delta$ over $S \in \mathcal{D}^n$, we have*

$$\mathbb{E}_{(x,y) \sim \mathcal{D}}[\mathbb{1}\{f_S(x) \neq y\}] \leq \inf_{h \in \mathcal{H}} \mathbb{E}_{(x,y) \sim \mathcal{D}}[\mathbb{1}\{h(x) \neq y\}] + 100\sqrt{\frac{k(n) \log \frac{n}{k(n)} + k(n) + \log \frac{1}{\delta}}{n}}.$$

## 2.3 Covering Numbers, Metric Entropy, and Complexity Measures

In Section 4, we provide conditions for which a hypothesis class $\mathcal{H}$ is online learnable under smoothed adversaries. While sufficient conditions for learnability are typically established via combinatorial

dimensions, our sufficient conditions will be in terms of covering/packing numbers of $\mathcal{H}$ using a distance metric that depends on the base measure $\mu$. This discrepancy with existing literature is due to a simple observation: any parameter of $\mathcal{H}$ alone cannot characterize smoothed online classification. Indeed, if one takes the base measure $\mu$ to be a Dirac measure, then every $\mathcal{H} \subseteq \mathcal{Y}^{\mathcal{X}}$ is trivially online learnable under a smoothed adversary. Accordingly, any meaningful characterization of smoothed online classification must be in terms of both $\mathcal{H}$ and $\mu$.

To start, we first define $\varepsilon$-covering numbers for generic metric spaces $(\mathcal{G}, d)$.

**Definition 2** (Covering Number). *Let $(\mathcal{G}, d)$ be a bounded metric space. A subset $\mathcal{G}' \subseteq \mathcal{G}$ is an $\varepsilon$-cover for $\mathcal{G}$ with respect to $d$ if for every $g \in \mathcal{G}$, there exists an $g' \in \mathcal{G}'$ such that $d(g, g') \leq \varepsilon$. The covering number of $\mathcal{G}$ at scale $\varepsilon$, denoted $\mathcal{N}(\varepsilon, \mathcal{G}, d)$, is the smallest $n \in \mathbb{N}$ such that there exists an $\varepsilon$-cover of $\mathcal{G}$ with cardinality $n$. That is, $\mathcal{N}(\varepsilon, \mathcal{G}, d) := \inf\{|\mathcal{G}'| : \mathcal{G}' \text{ is an } \varepsilon\text{-cover for } \mathcal{G}\}$.*

The metric entropy for $(\mathcal{G}, d)$ at scale $\varepsilon > 0$ is defined as $\log \mathcal{N}(\varepsilon, \mathcal{G}, d)$. In this paper, we consider the metric space $(\mathcal{H}, d_\mu)$ where $\mathcal{H} \subseteq \mathcal{Y}^{\mathcal{X}}$ is a hypothesis class and $d_\mu(h_1, h_2) = \mathbb{P}_{x \sim \mu}[h_1(x) \neq h_2(x)]$ for some $\mu \in \Pi(\mathcal{X})$. The key complexity measure in this work is

$$C_{\varepsilon, \sigma}(\mathcal{H}, \mu) := \sup_{n \in \mathbb{N}} \sup_{\nu_{1:n} \in \mathrm{B}(\mu, \sigma)} \mathbb{E}_{x_{1:n} \sim \nu_{1:n}} \left[ \mathcal{N}(\varepsilon, \mathcal{H}, d_{\hat{\mu}_n}) \right],$$

where $\hat{\mu}_n$ denotes the *empirical* measure over $x_{1:n}$. At a high-level, $C_{\varepsilon, \sigma}(\mathcal{H}, \mu)$ measures the complexity of $\mathcal{H}$ in terms of its average empirical covering number, where the average is taken over processes from $\mathrm{B}(\mu, \sigma)$. Using $C_{\varepsilon, \sigma}(\mathcal{H}, \mu)$, we define the property of uniformly bounded empirical metric entropy.

**Definition 3** (Uniformly Bounded Empirical Metric Entropy). *A hypothesis class $\mathcal{H} \subseteq \mathcal{Y}^{\mathcal{X}}$ has the Uniformly Bounded Empirical Metric Entropy* (UBEME) *property with respect to $\mu$ if $C_{\varepsilon, \sigma}(\mathcal{H}, \mu) < \infty$ for every $\varepsilon, \sigma > 0$.*

In Theorem 4.1, we show that $\mathcal{H}$ is online learnable under smoothed adversaries if $\mathcal{H}$ enjoys the UBEME property with respect to the base measure $\mu$.

## 3 PAC Learnability is Not Sufficient for Smoothed Online Learnability

In Section 4, we show that the PAC learnability of $\mathcal{H}$ is sufficient for its smoothed online learnability when $|\mathcal{Y}| < \infty$. Here, we show that this is not the case when $|\mathcal{Y}|$ is unbounded by constructing a PAC learnable hypothesis class that is not smoothed online learnable. In fact, we prove a *stronger* result.

**Theorem 3.1.** *There exists a hypothesis class $\mathcal{H} \subseteq \mathcal{Y}^{\mathcal{X}}$ such that following holds:*

*(i) $\mathcal{H}$ has a compression scheme of size $1$.*

*(ii) For $\mu = Uniform(\mathcal{X})$ and $\sigma = 1$, we have $\inf_{\mathcal{A}} \mathrm{R}_{\mathcal{A}}^{\mu, \sigma}(T, \mathcal{H}) \geq \frac{T}{2}$.*

The part (i) of Theorem 3.1, together with Theorem 2.1, shows that $\mathcal{H}$ is agnostic PAC learnable with error rate $\varepsilon(\delta, n) = O\left(\sqrt{\frac{\log n + \log(1/\delta)}{n}}\right)$. On the other hand, based on Definition 1, part (ii) shows that $\mathcal{H}$ is not smoothed online learnable. Together, we infer the following corollary.

**Corollary 3.2** (Agnostic PAC Learnability $\not\Rightarrow$ Smoothed Online Learnability). *There exists $\mathcal{H} \subseteq \mathcal{Y}^{\mathcal{X}}$ such that $\mathcal{H}$ is agnostic PAC learnable but not learnable in the smoothed setting under $\mu = Uniform(\mathcal{X})$ and $\sigma = 1$.*

When $|\mathcal{Y}| < \infty$, the existence of a $O(1)$-size compression schemes and agnostic PAC learnability are equivalent [David et al., 2016]. Thus, there is no qualitative difference between Theorem 3.1 and Corollary 3.2. However, when $|\mathcal{Y}| = \infty$, a recent work has shown that the multiclass PAC learnability does not imply the existence of $O(1)$-size compression schemes [Pabbaraju, 2024]. Thus, Theorem 3.1 is a qualitatively stronger result than Corollary 3.2. Moreover, our proof of Theorem 3.1 below provides an explicit PAC learner for $\mathcal{H}$.

*Proof.* (of Theorem 3.1) Let $\mathcal{X} = [0, 1]$. Given a bitstring $\theta = (\theta_1, \ldots, \theta_n) \in \{0, 1\}^n$, define $\theta_{\leq t} := (\theta_1, \ldots, \theta_t)$ and $\theta_{<t} := (\theta_1, \ldots, \theta_{t-1})$ for any $t \in \{1, \ldots, n\}$. Fix an $n \in \mathbb{N}$ and ordered

sequence $(x_1, x_2, \ldots, x_n) \in \mathcal{X}^n$ such that $x_i \neq x_j$ for all $i \neq j$. For every $\theta \in \{0,1\}^n$, define

$$h^\theta_{(x_1,\ldots,x_n)}(x) := \begin{cases} ((x_1,\ldots,x_n), \theta_{\leq t}) & \text{if } \exists t \in [n] \text{ such that } x = x_t \\ ((x_1,\ldots,x_n), \star) & \text{otherwise} \end{cases}$$

Let $O_n := \{(x_1, x_2, \ldots, x_n) \in \mathcal{X}^n \, : \, x_i \neq x_j\}$ be the set of all ordered sequences of length $n$ with distinct elements. Then, we define our hypothesis class to be

$$\mathcal{H} := \bigcup_{n \in \mathbb{N}} \; \bigcup_{(x_1,\ldots,x_n) \in O_n} \; \bigcup_{\theta \in \{0,1\}^n} \left\{ h^\theta_{(x_1,\ldots,x_n)} \right\}.$$

Here, the label space is $\mathcal{Y} := \cup_{h \in \mathcal{H}} \{\text{image}(h)\}$. For any $y \in \mathcal{Y}$, let us define $y[1]$ and $y[2]$ to be the first and the second entry of the tuple $y$ respectively. Note that $y[1] \in O_m$ for some $m \in \mathbb{N}$ and $y[2] \in \{0,1\}^t$ for some $t \leq m$.

**Proof of (i).** We now define a compression scheme $(\kappa, \rho)$ of size 1 for $\mathcal{H}$.

- Define a compression function $\kappa : \cup_{i \geq 1} (\mathcal{X} \times \mathcal{Y})^i \to \mathcal{X} \times \mathcal{Y}$ as follows. Given any realizable sample $S = \{(x_1, y_1), \ldots, (x_n, y_n)\}$ of size $n \in \mathbb{N}$, the function $\kappa$ outputs $\kappa(S) = \{(x_1, y_1)\}$ if $y_i[2] = \star$ for all $i \in [n]$. On the other hand, if there exists a $y_i$ such that $y_i[2] \in \{0,1\}^t$ for some $t \in \mathbb{N}$, then $\kappa(S) = \{(x_\ell, y_\ell)\}$. Here, $\ell \in [n]$ is the index such that $y_\ell[2]$ is the longest binary string among all $i \in [n]$ for which $y_i[2] \neq \star$.

- Define a reconstruction function $\rho : \mathcal{X} \times \mathcal{Y} \to \mathcal{H}$ as follows. Given an output of the compression function $\{(x, y)\}$, the reconstruction function outputs $\rho(\{(x, y)\}) = h^{\mathbf{0}}_{y[1]} \in \mathcal{H}$ if $y[2] = \star$. Here, $\mathbf{0}$ is all 0's bitstring of length equal to that of the tuple $y[1]$. On the other hand, if $y[2] \in \{0,1\}^t$ for some $t \leq |y[1]|$, then output $\rho(\{(x, y)\}) = h^\theta_{y[1]} \in \mathcal{H}$, where $\theta$ is an arbitrary bitstring of length $y[1]$ such that $\theta_{\leq t} = y[2]$.

Next, we show that $(\kappa, \rho)$ is a valid compression scheme for $\mathcal{H}$. Let $S = \{(x_1, y_1), \ldots, (x_n, y_n)\} \in (\mathcal{X} \times \mathcal{Y})^n$ denote any sample of size $n$ that is realizable by $\mathcal{H}$. We want to show that $f = \rho(\kappa(S))$ satisfies $f(x_i) = y_i$ for all $i \in [n]$. Since $S$ is realizable by $\mathcal{H}$, there exists $m \in \mathbb{N}$, $(z_1, \ldots, z_m) \in \mathcal{X}^m$ such that $z_i \neq z_j$ for all $i, j \in [m]$ and $\beta \in \{0,1\}^m$ such that $h^\beta_{(z_1,\ldots,z_m)}(x_i) = y_i$ for all $i \in [n]$. By definition of $\mathcal{H}$, for all $i \in [n]$, we have $y_i[1] = (z_1, \ldots, z_m)$ and $y_i[2] \in \{\star, \beta_{\leq t}\}$ for some $t \leq m$. Given a realizable sample $S$, there are two cases to consider: (a) $y_i[2] = \star$ for all $i \in [n]$ and (b) there exists $i \in [n]$ such that $y_i[2] \neq \star$.

If we are in case (a), then we know that $x_i \notin \{z_1, \ldots, z_m\}$ for all $i \in [n]$. Moreover, we have $\kappa(S) = \{(x_1, y_1)\}$ where $y_1[1] = (z_1, \ldots, z_m)$ and $y_1[2] = \star$. By definition of the reconstruction function, $\rho(\{(x_1, y_1)\}) = h^{\mathbf{0}}_{(z_1,\ldots,z_m)}$. Since $x_i \notin \{z_1, \ldots, z_m\}$, by definition of $h^{\mathbf{0}}_{(z_1,\ldots,z_m)}$, we have $h^{\mathbf{0}}_{(z_1,\ldots,z_m)}(x_i) = ((z_1, \ldots, z_m), \star) = y_i$ for all $i \in [n]$. Thus, $(\kappa, \rho)$ is a valid compression scheme for $\mathcal{H}$ in this case.

Suppose (b) is true. Define $I_S := \{i \in [n] : y_i[2] \neq \star\}$. Since $h^\beta_{(z_1,\ldots,z_m)}$ is consistent with the sample $S$, we must have $y_i[2] = \beta_{\leq |y_i[2]|}$ for each $i \in I_S$. Here, $|y_i[2]|$ is the length of bitstring $y_i[2]$. Let $\ell \in I_S$ such that $|y_\ell[2]| \geq |y_i[2]|$ for all $i \in I_S$. By definition of the compression function $\kappa$, we have $\kappa(S) = \{(x_\ell, y_\ell)\}$, where $y_\ell[1] = (z_1, \ldots, z_m)$ and $y_\ell[2] = \beta_{\leq |y_\ell[2]|}$. Let $\theta \in \{0,1\}^m$ be the completion of $y_\ell[2]$ such that the reconstruction function returns $\rho(\{(x_\ell, y_\ell)\}) = h^\theta_{(z_1,\ldots,z_m)}$. By definition of $\rho$, we have $\beta_{\leq t} = \theta_{\leq t}$ for $t = |y_\ell[2]|$. To complete our proof, it suffices to show that $h^\theta_{(z_1,\ldots,z_m)}(x_i) = y_i$ for all $i \in [n]$. There are two cases to consider: $i \in I_S$ and $i \notin I_S$. When $i \notin I_S$, we have $x_i \notin \{z_1, \ldots, z_m\}$ and thus $h^\theta_{(z_1,\ldots,z_m)}(x_i) = ((z_1, \ldots, z_m), \star) = y_i$. As for the index $i \in I_S$, we must have that $x_i \in \{z_1, \ldots, z_m\}$. Otherwise, $y_i[2]$ would be equal to $\star$, contradicting the fact that $i \in I_S$. In fact, by definition of $h^\beta_{(z_1,\ldots,z_m)}$, we have $x_i = z_{|y_i[2]|}$. This implies that, for all $i \in I_S$, we have $h^\theta_{(z_1,\ldots,z_m)}(x_i) = h^\theta_{(z_1,\ldots,z_m)}(z_{|y_i[2]|}) = ((z_1, \ldots, z_m), \theta_{\leq |y_i[2]|}) = ((z_1, \ldots, z_m), \beta_{\leq |y_i[2]|}) = y_i$. Here, we use that $|y_i[2]| \leq |y_\ell[2]|$ for all $i \in I_S$ and the fact that $\theta_{\leq t} = \beta_{\leq t}$ for all $t \leq |y_\ell[2]|$. Therefore, we have shown that $h^\theta_{(z_1,\ldots,z_m)}(x_i) = y_i$ for all $i \in [n]$.

**Proof of (ii).** Let $\mu = \text{Uniform}([0,1])$. We now specify the stream $\{(x_t, y_t)\}_{t=1}^T$ to be observed by the learner. For the instances, we take $x_1, \ldots, x_T \sim \mu$ to be iid samples from $\mu$. Note that $x_1, \ldots, x_T$

are distinct with probability 1. Moreover, as all the instances are drawn from the same distribution $\mu$, this adversary is $\sigma$-smooth for $\sigma = 1$. To specify $y_1, \ldots, y_T$, we first draw $\theta \sim \text{Uniform}(\{0,1\}^T)$ and define $y_t = ((x_1, \ldots, x_T), \theta_{\leq t})$ for all $t \in [T]$. Given distinct $x_1, \ldots, x_T$, for any algorithm $\mathcal{A}$, we first show that

$$\mathop{\mathbb{E}}_{\theta \sim \text{Uniform}(\{0,1\}^T)} \left( \sum_{t=1}^{T} \mathbb{E}_{\mathcal{A}} \left[ \mathbb{1}\{\mathcal{A}(x_t) \neq y_t\} \right] - \inf_{h \in \mathcal{H}} \sum_{t=1}^{T} \mathbb{1}\{h(x_t) \neq y_t\} \right) \geq \frac{T}{2}. \tag{1}$$

The probabilistic method implies the existence of a $\theta \in \{0,1\}^T$ such that the claimed bound of $T/2$ holds. This subsequently implies that, for a distinct $x_1, \ldots, x_T$, we have

$$\sup_{y_{1:T}} \left( \sum_{t=1}^{T} \mathbb{E}_{\mathcal{A}} \left[ \mathbb{1}\{\mathcal{A}(x_t) \neq y_t\} \right] - \inf_{h \in \mathcal{H}} \sum_{t=1}^{T} \mathbb{1}\{h(x_t) \neq y_t\} \right) \geq \frac{T}{2}.$$

Finally, using the fact that $x_1, \ldots, x_T \sim \mu$ are distinct with probability 1, we obtain the bound

$$\inf_{\mathcal{A}} R_{\mathcal{A}}^{\mu, \sigma}(T, \mathcal{H}) \geq \mathop{\mathbb{E}}_{x_1, \ldots, x_T \sim \mu} \left[ \sup_{y_{1:T}} \left( \sum_{t=1}^{T} \mathbb{E}_{\mathcal{A}} \left[ \mathbb{1}\{\mathcal{A}(x_t) \neq y_t\} \right] - \inf_{h \in \mathcal{H}} \sum_{t=1}^{T} \mathbb{1}\{h(x_t) \neq y_t\} \right) \right] \geq \frac{T}{2}.$$

To complete our proof, it now suffices to prove Equation (1). Fixing distinct $x_1, \ldots, x_T$, the lowerbound on the expected cumulative loss of the algorithm $\mathcal{A}$ is

$$\sum_{t=1}^{T} \mathop{\mathbb{E}}_{\theta \sim \text{Unif}(\{0,1\}^T)} \left[ \mathbb{E}_{\mathcal{A}} \left[ \mathbb{1}\{\mathcal{A}(x_t) \neq y_t\} \right] \right] = \sum_{t=1}^{T} \mathbb{E} \left[ \mathbb{E}_{\theta_t} \left[ \mathbb{1}\{\mathcal{A}(x_t) \neq ((x_1, \ldots, x_T), \theta_{\leq t})\} \right] \mid \mathcal{A}, \theta_{<t} \right]$$

$$= \sum_{t=1}^{T} \mathbb{E} \left[ \frac{1}{2} \mathbb{1} \left\{ \mathcal{A}(x_t) \neq ((x_{1:T}), (\theta_{<t}, 0)) \right\} + \frac{1}{2} \mathbb{1} \left\{ \mathcal{A}(x_t) \neq (x_{1:T}, (\theta_{<t}, 1)) \right\} \right]$$

$$\geq \sum_{t=1}^{T} \frac{1}{2} = \frac{T}{2}.$$

At a high-level, we use the fact that $\theta_t$ is sampled uniformly at random from $\{0, 1\}$ and is independent of $\mathcal{A}$ as well as $\theta_i$ for all $i \neq t$. Thus, on each round, the algorithm cannot do any better than randomly guessing the value of $\theta_t$. Next, we upperbound the expected loss of the best-fixed function in hindsight. Given distinct $x_1, \ldots, x_T$ and $\theta \in \{0, 1\}^T$, we can pick the hypothesis $h_{(x_1, \ldots, x_T)}^{\theta}$. By definition of this hypothesis, we have $h_{(x_1, \ldots, x_T)}^{\theta}(x_t) = ((x_1, \ldots, x_T), \theta_{\leq t}) = y_t$. Thus, for every distinct $x_1, \ldots, x_T$, we have

$$\mathop{\mathbb{E}}_{\theta \sim \text{Unif}(\{0,1\}^T)} \left[ \inf_{h \in \mathcal{H}} \sum_{t=1}^{T} \mathbb{1}\{h(x_t) \neq y_t\} \right] \leq \mathop{\mathbb{E}}_{\theta \sim \text{Unif}(\{0,1\}^T)} \left[ \sum_{t=1}^{T} \mathbb{1}\{h_{(x_1, \ldots, x_T)}^{\theta}(x_t) \neq y_t\} \right] = 0.$$

Finally, equation (1) follows upon combining the lowerbound on the cumulative loss of $\mathcal{A}$ and the upperbound on the cumulative loss of the optimal hypothesis in hindsight. $\square$

To prove the qualitative separation between PAC and smoothed online learnability in Theorem 3.1, we required $|\mathcal{Y}|$ to be unbounded. The following theorem, proved in Appendix C, shows the quantitative dependence of the regret on $|\mathcal{Y}|$ when it is bounded.

**Theorem 3.3.** *For every $K \in \mathbb{N}$, there exists $\mathcal{H} \subseteq \mathcal{Y}^{\mathcal{X}}$ with $|\mathcal{Y}| = K$ such that $\mathcal{H}$ has a compression scheme of 1, but $\inf_{\mathcal{A}} R_{\mathcal{A}}^{\mu, \sigma}(T, \mathcal{H}) \geq \frac{\log K}{24 \log \log K}$ for $\mu = Uniform(\mathcal{X})$ and $\sigma = 1$.*

Theorem 3.3 shows that one can get quantitative separation whenever $K \geq 2^{T \log T}$.

## 4 A Sufficient Condition for Smoothed Online Classification

In this section, we provide a sufficient condition for smoothed online classification. Our main result provides a quantitative upperbound on the expected regret in terms of $C_{\varepsilon, \sigma}(\mathcal{H}, \mu)$.

**Theorem 4.1.** *For every $\mathcal{H} \subseteq \mathcal{Y}^{\mathcal{X}}$, $\mu \in \Pi(\mathcal{X})$ and $\sigma > 0$, we have that*

$$\inf_{\mathcal{A}} \mathrm{R}_{\mathcal{A}}^{\mu,\sigma}(T, \mathcal{H}) \le 6 \inf_{\varepsilon > 0} \left\{ \frac{\varepsilon T}{\sigma} + \sqrt{T \log(C_{\varepsilon^2, \sigma}(\mathcal{H}, \mu))} \right\}.$$

Theorem 4.1 shows that as long as $\mathcal{H}$ satisfies the UBEME condition with respect to $\mu$, it is online learnable under a smoothed adversary. As a corollary, we also establish the following sufficient condition in terms of the Graph dimension, a combinatorial dimension characterizing PAC learnability when $|\mathcal{Y}| < \infty$ (see Appendix A for a complete definition) [Natarajan, 1989].

**Corollary 4.2.** *For every $\mathcal{H} \subseteq \mathcal{Y}^{\mathcal{X}}$, $\mu \in \Pi(\mathcal{X})$ and $\sigma > 0$, we have that*

$$\inf_{\mathcal{A}} \mathrm{R}_{\mathcal{A}}^{\mu,\sigma}(T, \mathcal{H}) \le 6 \inf_{\varepsilon > 0} \left\{ \frac{\varepsilon T}{\sigma} + \sqrt{T \, \mathrm{G}(\mathcal{H}) \log\left(\frac{41|\mathcal{Y}|}{\varepsilon^2}\right)} \right\} \le 12 \sqrt{T \, \mathrm{G}(\mathcal{H}) \log\left(\frac{41 \, T \, |\mathcal{Y}|}{\sigma^2}\right)},$$

*where $\mathrm{G}(\mathcal{H})$ denotes the Graph dimension of $\mathcal{H}$.*

Corollary 4.2, whose proof is in Appendix E, shows that PAC learnability of $\mathcal{H}$ is sufficient for smoothed online classification whenever $|\mathcal{Y}| < \infty$. When $|\mathcal{Y}| = 2$, this bounds, up to a constant factor, recovers that from Haghtalab [2018]. We now proceed with the proof of Theorem 4.1.

*Proof.* (of Theorem 4.1) Let $\nu_1, \dots, \nu_T \in \mathrm{B}(\mu, \sigma)$ denote the sequence of $\sigma$-smooth distributions picked by the adversary. Fix a $\varepsilon > 0$. Then, by Lemma B.2, we have that $\mathcal{N}(\varepsilon, \mathcal{H}, d_\mu) \le C_{\frac{\varepsilon}{2}, \sigma}(\mathcal{H}, \mu)$. Let $\mathcal{H}' \subset \mathcal{H}$ denote an $\varepsilon$-cover with respect to $d_\mu$ of size at most $C_{\frac{\varepsilon}{2}, \sigma}(\mathcal{H}, \mu)$. Let $\mathcal{A}$ denote the online learner that runs the Randomized Exponential Weights Algorithm (REWA) on the the data stream $(x_1, y_1), ..., (x_T, y_T)$ using $\mathcal{H}'$ as its set of experts. By the guarantees of REWA,

$$\mathbb{E}_{\mathcal{A}}\left[ \sum_{t=1}^T \mathbb{1}\{\mathcal{A}(x_t) \ne y_t\} \right] \le \inf_{h' \in \mathcal{H}'} \sum_{t=1}^T \mathbb{1}\{h'(x_t) \ne y_t\} + \sqrt{2T \log(|\mathcal{H}'|)}$$

$$\le \inf_{h' \in \mathcal{H}'} \sum_{t=1}^T \mathbb{1}\{h'(x_t) \ne y_t\} + \sqrt{2T \log(C_{\frac{\varepsilon}{2}, \sigma}(\mathcal{H}, \mu))}$$

$$\le \inf_{h \in \mathcal{H}} \sum_{t=1}^T \mathbb{1}\{h(x_t) \ne y_t\} + \sup_{h \in \mathcal{H}} \inf_{h' \in \mathcal{H}'} \sum_{t=1}^T \mathbb{1}\{h'(x_t) \ne h(x_t)\} + \sqrt{2T \log(C_{\frac{\varepsilon}{2}, \sigma}(\mathcal{H}, \mu))}$$

where the expectation is only taken with respect to the randomness of the REWA and the last inequality follows by the triangle inequality. Taking an outer expectation with respect to the process $x_{1:T} \sim \nu_{1:T}$, we have $\mathbb{E}\left[ \sum_{t=1}^T \mathbb{1}\{\mathcal{A}(x_t) \ne y_t\} \right]$ is at most

$$\mathbb{E}_{x_{1:T} \sim \nu_{1:T}}\left[ \inf_{h \in \mathcal{H}} \sum_{t=1}^T \mathbb{1}\{h(x_t) \ne y_t\} \right] + \mathbb{E}_{x_{1:T} \sim \nu_{1:T}}\left[ \sup_{h \in \mathcal{H}} \inf_{h' \in \mathcal{H}'} \sum_{t=1}^T \mathbb{1}\{h'(x_t) \ne h(x_t)\} \right] + \sqrt{2T \log(C_{\frac{\varepsilon}{2}, \sigma}(\mathcal{H}, \mu))}.$$

It remains to bound $\mathbb{E}\left[ \sup_{h \in \mathcal{H}} \inf_{h' \in \mathcal{H}'} \sum_{t=1}^T \mathbb{1}\{h'(x_t) \ne h(x_t)\} \right]$. We provide a sketch of the proof here and defer the full details to Appendix D. Consider the class $\mathcal{G} = \{x \mapsto \mathbb{1}\{h'(x) \ne h(x)\} : h \in \mathcal{H}\}$, where $h' \in \mathcal{H}'$ denotes the $\varepsilon$-cover of $h$ with respect to $d_\mu$, and note that

$$\mathbb{E}_{x_{1:T} \sim \nu_{1:T}}\left[ \sup_{h \in \mathcal{H}} \inf_{h' \in \mathcal{H}'} \sum_{t=1}^T \mathbb{1}\{h'(x_t) \ne h(x_t)\} \right] \le \mathbb{E}_{x_{1:T} \sim \nu_{1:T}}\left[ \sup_{g \in \mathcal{G}} \sum_{t=1}^T g(x_t) \right].$$

By standard symmetrization arguments, we get

$$\mathbb{E}_{x_{1:T} \sim \nu_{1:T}}\left[ \sup_{g \in \mathcal{G}} \sum_{t=1}^T g(x_t) \right] \le \sup_{g \in \mathcal{G}} \mathbb{E}_{x_{1:T} \sim \nu_{1:T}}\left[ \sum_{t=1}^T g(x_t') \right] + 2T \mathbb{E}_{x_{1:T} \sim \nu_{1:T}}\left[ \hat{\mathfrak{R}}(\mathcal{G}, x_{1:T}) \right]$$

where $\hat{\mathfrak{R}}(\mathcal{G}, x_{1:T})$ is the Rademacher complexity of $\mathcal{G}$ (see Appendix A). Note that $\mathcal{G} \subseteq \mathcal{H} \Delta \mathcal{H}$ where $\mathcal{H} \Delta \mathcal{H} := \{x \mapsto \mathbb{1}\{h_1(x) \ne h_2(x)\} : h_1, h_2 \in \mathcal{H}\}$, and thus $\hat{\mathfrak{R}}(\mathcal{G}, x_{1:T}) \le \hat{\mathfrak{R}}(\mathcal{H} \Delta \mathcal{H}, x_{1:T})$. Using

the discretization-based upperbound (Lemma A.1) on the empirical Rademacher complexity and a relation between the covering numbers of $\mathcal{H}$ and $\mathcal{H}\Delta\mathcal{H}$ (Lemma B.3), we have

$$\hat{\mathfrak{R}}(\mathcal{H}\Delta\mathcal{H}, x_{1:T}) \leq \varepsilon + \sqrt{\frac{2\log\mathcal{N}(\varepsilon, \mathcal{H}\Delta\mathcal{H}, \rho_{\hat{\mu}_T})}{T}} \leq \varepsilon + \sqrt{\frac{2\log\mathcal{N}(\varepsilon^2, \mathcal{H}\Delta\mathcal{H}, d_{\hat{\mu}_T})}{T}} \leq \varepsilon + 2\sqrt{\frac{\log\mathcal{N}(\frac{\varepsilon^2}{2}, \mathcal{H}, d_{\hat{\mu}_T})}{T}}.$$

Plugging in the upperbound on the Rademacher complexity and using the change of measure, $\sigma$-smoothness, and the definition of $\mathcal{H}'$ on the first term gives

$$\mathbb{E}_{x_{1:T}\sim\nu_{1:T}}\left[\sup_{g\in\mathcal{G}}\sum_{t=1}^{T} g(x_t)\right] \leq \frac{\varepsilon T}{\sigma} + 2\varepsilon T + 4\sqrt{T\log\left(\mathbb{E}_{x_{1:T}\sim\nu_{1:T}}\left[\mathcal{N}\left(\frac{\varepsilon^2}{2}, \mathcal{H}, d_{\hat{\mu}_T}\right)\right]\right)}.$$

Using the definition of $C_{\varepsilon,\sigma}(\mathcal{H}, \mu)$ to get

$$\mathbb{E}_{x_{1:T}\sim\nu_{1:T}}\left[\sup_{h\in\mathcal{H}}\inf_{h\in\mathcal{H}'}\sum_{t=1}^{T}\mathbb{1}\{h'(x_t)\neq h(x_t)\}\right] \leq \frac{\varepsilon T}{\sigma} + 2\varepsilon T + 4\sqrt{T\log C_{\frac{\varepsilon^2}{2},\sigma}(\mathcal{H}, \mu)},$$

substituting into the regret bound for $\mathcal{A}$, and doing some algebra completes the proof sketch. $\square$

Our upperbounds in terms of expected empirical covering numbers can be meaningful even when VC and Graph dimension of $\mathcal{H}$ is infinity. As a simple example, let $\mathcal{X} = [0,1]$, $\mu = \text{Uniform}(\mathcal{X})$, and consider the binary hypothesis class $\mathcal{H} = \{x \mapsto \mathbb{1}\{x \in A\} : A \subset \mathbb{Q}, |A| < \infty\}$. It's not too hard to see that $\text{VC}(\mathcal{H}) = \infty$. On the other hand, $C_{\varepsilon,\sigma}(\mathcal{H}, \mu) = 1$ since for every $n \in \mathbb{N}$, the sample $x_{1:n} \sim \mu$ does not lie in $\mathbb{Q}$ almost surely, and when $x_{1:n} \notin \mathbb{Q}$, $d_{\hat{\mu}_n}(h_1, h_2) = 0$ for all $h_1, h_2 \in \mathcal{H}$.

To prove Theorem 4.1, we show that the UBEME implies a bound on the metric entropy $\mathcal{N}(\varepsilon, \mathcal{H}, d_\mu)$. It is natural ask whether the finiteness of $\mathcal{N}(\varepsilon, \mathcal{H}, d_\mu)$ for every $\varepsilon > 0$ alone is necessary and sufficient for smoothed online learnability. Unfortunately, it is neither sufficient nor necessary.

**Theorem 4.3.** *Let $\mathcal{X} = [0,1]$ and $\mu = \text{Uniform}(\mathcal{X})$. Then,*

(i) *There exists a $\mathcal{H} \subseteq \{0,1\}^{\mathcal{X}}$ such that $\mathcal{N}(\varepsilon, \mathcal{H}, d_\mu) = 1$ for every $\varepsilon > 0$ but $\text{R}_{\mathcal{A}}^{\mu,\sigma}(T, \mathcal{H}) \geq \frac{T}{2}$ for every $\sigma > 0$.*

(ii) *There exists $\mathcal{H} \subseteq \mathbb{N}^{\mathcal{X}}$ such that $\mathcal{N}(\varepsilon, \mathcal{H}, d_\mu) = \infty$ for every $\varepsilon > 0$ but $\inf_{\mathcal{A}} \text{R}_{\mathcal{A}}^{\mu,\sigma}(T, \mathcal{H}) = O(\sqrt{T\log(T)})$ for every $\sigma > 0$.*

*Proof.* We first prove (i). Consider the hypothesis class $\mathcal{H} = \{x \mapsto \mathbb{1}\{x \in S\} : S \subset \mathcal{X}, |S| < \infty\}$. Note that for every $h_1, h_2 \in \mathcal{H}$, we have that $d_\mu(h_1, h_2) = \mathbb{P}_{x\sim\mu}[h_1(x) \neq h_2(x)] = 0$ since $h_1$ and $h_2$ disagree on at most a finite number of points in $\mathcal{X}$. Thus, for every $\varepsilon > 0$, $\mathcal{H}$ is trivially coverable using exactly one hypothesis in $\mathcal{H}$. To show that $\text{R}_{\mathcal{A}}^{\mu,\sigma}(T, \mathcal{H}) \geq \frac{T}{2}$ for every $\sigma > 0$, consider the adversary that picks $\nu_t = \mu$ for all $t \in [T]$. The process $x_{1:T} \sim \nu_{1:T}$ is then an iid draw from $\mu$ of length $T$. Consider the data stream $(x_1, y_1), ..., (x_T, y_T)$ where $x_{1:T} \sim \mu^T$ and $y_t \sim \text{Unif}(\{0, 1\})$ for every $t \in [T]$. Such a stream is realizable by $\mathcal{H}$ almost surely since the the sequence of instances $x_{1:T}$ are all distinct with probability 1 and there can be at most a finite number of timepoints where $y_t = 1$. On the other hand, any learning algorithm $\mathcal{A}$ must make at least $T/2$ mistakes in expectation (with respect to all sources of randomness), since the labels $y_t \sim \text{Unif}(\{0, 1\})$. Thus, by the probabilistic method, for every learning algorithm $\mathcal{A}$, there must exist a sequence of labels $y_{1:T}$, such that $\mathcal{A}$'s expected regret is $T/2$.

We now prove (ii). Let $\mathcal{H} = \{x \mapsto a : a \in \mathbb{N}\}$ be the class of constant functions. Note that for every $h_1, h_2 \in \mathcal{H}$, we have that $d_\mu(h_1, h_2) = 1$ since $h_1$ and $h_2$ disagree everywhere on $\mathcal{X}$. Thus, for every $\varepsilon < 1$, we have that $\mathcal{N}(\varepsilon, \mathcal{H}, d_\mu) = \infty$ since $|\mathcal{H}| = \infty$. On the other hand, the Littlestone dimension of $\mathcal{H}$ is 1. Thus, by Theorem 4 from Hanneke et al. [2023], we get that $\inf_{\mathcal{A}} \text{R}_{\mathcal{A}}^{\mu,\sigma}(T, \mathcal{H}) = O(\sqrt{T\log T})$ for every $\sigma > 0$. $\square$

## 5 Discussion

In this work, we show a separation between the learnability of $\mathcal{H} \subseteq \mathcal{Y}^{\mathcal{X}}$ in the PAC setting and the smoothed online setting when $|\mathcal{Y}|$ is unbounded. We also provide a sufficient condition for smoothed

online learnability under any base measure $\mu$. However, as noted in Section 4, our sufficient condition is not necessary for the smoothed learnability of $\mathcal{H}$. Thus, an important open question is to find a condition that is both necessary and sufficient for smoothed online learnability. Traditionally, in learning theory, learnability is characterized in terms of a combinatorial property of just the hypothesis class $\mathcal{H} \subseteq \mathcal{Y}^{\mathcal{X}}$. However, the property of $\mathcal{H}$ alone cannot provide a characterization of learnability in the smoothed online setting. Choosing $\mu$ to be a Dirac distribution will make any $\mathcal{H}$, even the set of all measurable functions from $\mathcal{X}$ to $\mathcal{Y}$, trivially learnable. Thus, the characterization of learnability must necessarily be a property of the tuple $(\mathcal{H}, \mu)$. To that end, we pose the following question.

Given $(\mathcal{H}, \mu)$, is there a complexity measure that characterizes the smoothed online learnability of $\mathcal{H}$ with base measure $\mu$?

## Acknowledgments and Disclosure of Funding

VR acknowledges the support from the NSF Graduate Research Fellowship Program.

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

## A  PAC Learnability, Combinatorial Dimensions, Complexity Measures

We begin by defining the agnostic PAC framework, a canonical learning model in the batch setting.

**Definition 4** (Agnostic PAC Learnability). *A hypothesis class $\mathcal{H} \subseteq \mathcal{Y}^{\mathcal{X}}$ is agnostic PAC learnable if there exists a function $m : (0,1)^2 \to \mathbb{N}$ and a learning algorithm $\mathcal{A} : \cup_{i \geq 1}(\mathcal{X} \times \mathcal{Y})^i \to \mathcal{Y}^{\mathcal{X}}$ with the following property: for every $\varepsilon, \delta \in (0,1)$ and for every distribution $\mathcal{D}$ on $\mathcal{X} \times \mathcal{Y}$, $\mathcal{A}$ running on $n \geq m(\varepsilon, \delta)$ i.i.d. samples from $\mathcal{D}$ outputs a predictor $\mathcal{A}_S$ such that with probability at least $1 - \delta$ over $S \sim \mathcal{D}^n$,*

$$\mathbb{E}_{(x,y)\sim\mathcal{D}} \left[ \mathbb{1}\{\mathcal{A}_S(x) \neq y\} \right] \leq \inf_{h \in \mathcal{H}} \mathbb{E}_{(x,y)\sim\mathcal{D}} \left[ \mathbb{1}\{h(x) \neq y\} \right] + \varepsilon.$$

The VC and Graph dimension characterize PAC learnability when $|\mathcal{Y}| = 2$ and $|\mathcal{Y}| < \infty$ respectively.

**Definition 5** (Vapnik-Chervonenkis (VC) Dimension). *A set $S = \{x_1, \ldots, x_d\} \subset \mathcal{X}$ is shattered by a binary function class $\mathcal{H} \subseteq \{0,1\}^{\mathcal{X}}$ if for every $\tau \in \{0,1\}^d$, there exists a hypothesis $h_\tau \in \mathcal{H}$ such that for all $i \in [d]$, we have $h_\tau(x_i) = \tau_i$. The VC dimension of $\mathcal{H}$, denoted $\mathrm{VC}(\mathcal{H})$, is the size of the largest shattered set $S \subseteq \mathcal{X}$. If $\mathcal{H}$ can shatter arbitrarily large sets, we say that $\mathrm{VC}(\mathcal{H}) = \infty$.*

**Definition 6** (Graph Dimension). *For any multiclass hypothesis class $\mathcal{H} \subseteq \mathcal{Y}^{\mathcal{X}}$, let $\ell \circ \mathcal{H} := \{(x,y) \mapsto \mathbb{1}\{h(x) \neq y\} : h \in \mathcal{H}\} \subseteq \{0,1\}^{(\mathcal{X} \times \mathcal{Y})}$ denotes its loss class. The Graph dimension of $\mathcal{H}$ is defined as $\mathrm{G}(\mathcal{H}) := \mathrm{VC}(\ell \circ \mathcal{H})$.*

Our proof of this result relies on bounding the Rademacher complexity, a canonical complexity measure used to establish generalization bounds in the batch setting [Bartlett and Mendelson, 2002].

**Definition 7** (Empirical Rademacher Complexity). *Let $\mathcal{F} \subseteq \mathbb{R}^{\mathcal{X}}$ and $x_1, ..., x_n \in \mathcal{X}^n$. The empirical Rademacher complexity of $\mathcal{F}$ with respect to $x_{1:n}$ is defined as*

$$\hat{\mathfrak{R}}(\mathcal{F}, x_{1:n}) = \frac{1}{n} \mathbb{E}_\tau \left[ \sup_{f \in \mathcal{F}} \left( \sum_{i=1}^n \tau_i f(x_i) \right) \right]$$

*where $\tau_1, ..., \tau_n$ are independent* Rademacher *random variables.*

The following upperbound on the empirical Rademacher complexity follows by a simple application of Massart's lemma [Shalev-Shwartz and Ben-David, 2014, Lemma 28.6].

**Lemma A.1** (Discretization Bound). *For every $\mathcal{F} \subseteq \mathbb{R}^{\mathcal{X}}$ and $x_1, ..., x_n \in \mathcal{X}^n$, we have that*

$$\hat{\mathfrak{R}}(\mathcal{F}, x_{1:n}) \leq \inf_{\varepsilon > 0} \left\{ \varepsilon + \left( \sup_{f \in \mathcal{F}} \sqrt{\mathop{\mathbb{E}}_{\hat{\mu}_n} [f^2]} \right) \sqrt{\frac{2 \log \mathcal{N}(\varepsilon, \mathcal{F}, \rho_{\hat{\mu}_n})}{n}} \right\}$$

*where $\hat{\mu}_n$ denotes the empirical measure on the sample $x_{1:n}$ and $\rho_{\mu}$ is the distance metric defined as*

$$\rho_{\hat{\mu}_n}(f_1, f_2) := \sqrt{\mathop{\mathbb{E}}_{x \sim \hat{\mu}_n} [(f_1(x) - f_2(x))^2]}.$$

Finally, we define the packing number for generic metric spaces.

**Definition 8** (Packing Number). *Let $(\mathcal{G}, d)$ be a bounded metric space. A subset $\mathcal{G}' \subseteq \mathcal{G}$ is an $\varepsilon$-packing with respect to $d$ if for every $g_i, g_j \in \mathcal{G}'$ we have that $d(g_i, g_j) > \varepsilon$. The packing number of $\mathcal{G}$ at scale $\varepsilon$, denoted $\mathcal{M}(\varepsilon, \mathcal{G}, d)$, is the largest $n \in \mathbb{N}$ such that there exists an $\varepsilon$-packing of $\mathcal{G}$ with cardinality $n$. That is, $\mathcal{M}(\varepsilon, \mathcal{G}, d) := \sup\{|\mathcal{G}'| : \mathcal{G}' \text{ is an } \varepsilon\text{-packing for } \mathcal{G}\}$.*

# B   Helper Lemmas

**Lemma B.1** (Covering-Packing duality [Anthony and Bartlett, 1999]). *For any metric space $(\mathcal{G}, d)$ and $\varepsilon > 0$, we have that*

$$\mathcal{M}(2\varepsilon, \mathcal{G}, d) \leq \mathcal{N}(\varepsilon, \mathcal{G}, d) \leq \mathcal{M}(\varepsilon, \mathcal{G}, d).$$

Using the Covering-Packing duality, we prove the following technical Lemma.

**Lemma B.2** (UBEME $\implies$ Bounded Metric Entropy with respect to $\mu$). *For any $\mu \in \Pi(\mathcal{X})$, hypothesis class $\mathcal{H} \subseteq \mathcal{Y}^{\mathcal{X}}$, and $\varepsilon > 0$, we have that*

$$\mathcal{N}(\varepsilon, \mathcal{H}, d_\mu) \leq \sup_{m \in \mathbb{N}} \mathop{\mathbb{E}}_{x_{1:m} \sim \mu^m} \left[ \mathcal{N}\left( \frac{\varepsilon}{2}, \mathcal{H}, d_{\hat{\mu}_m} \right) \right].$$

*Proof.* Fix $\varepsilon > 0$. By the Covering-Packing duality, we have that

$$\sup_{m \in \mathbb{N}} \mathop{\mathbb{E}}_{x_{1:m} \sim \mu^m} [\mathcal{M}(\varepsilon, \mathcal{H}, d_{\hat{\mu}_m})] \leq \sup_{m \in \mathbb{N}} \mathop{\mathbb{E}}_{x_{1:m} \sim \mu^m} \left[ \mathcal{N}\left( \frac{\varepsilon}{2}, \mathcal{H}, d_{\hat{\mu}_m} \right) \right].$$

Thus, it suffices to show that $\mathcal{N}(\varepsilon, \mathcal{H}, d_\mu) \leq \sup_{m \in \mathbb{N}} \mathbb{E}_{x_{1:m} \sim \mu^m} [\mathcal{M}(\varepsilon, \mathcal{H}, d_{\hat{\mu}_m})]$. Suppose, for the sake of contradiction, we have that $\mathcal{N}(\varepsilon, \mathcal{H}, d_\mu) > \sup_{m \in \mathbb{N}} \mathbb{E}_{x_{1:m} \sim \mu^m} [\mathcal{M}(\varepsilon, \mathcal{H}, d_{\hat{\mu}_m})]$. Then by the Covering-Packing duality, we have that $\mathcal{M}(\varepsilon, \mathcal{H}, d_\mu) > \sup_{m \in \mathbb{N}} \mathbb{E}_{x_{1:m} \sim \mu^m} [\mathcal{M}(\varepsilon, \mathcal{H}, d_{\hat{\mu}_m})] =: c$. By the definition of $\varepsilon$-packing, we can find $n > c$ hypothesis $h_1, ..., h_n \in \mathcal{H}$ and $\delta > 0$ such that $d_\mu(h_i, h_j) > \varepsilon + \delta$ for every $i \neq j$.

Fix $m \in \mathbb{N}$ and consider a sample $x_{1:m} \sim \mu^m$. Let $S_{ij} = \sum_{t=1}^m \mathbb{1}\{h_i(x_t) \neq h_j(x_t)\}$ denote the random variable counting the number of samples on which $h_i$ and $h_j$ differ. Note that $S_{ij} \sim \text{Binom}(m, d_\mu(h_i, h_j))$. Let $E_m$ be the event that $S_{ij} > \varepsilon m$ for all $i < j$. Then,

$$\begin{aligned}
\mathbb{P}_{x_{1:m} \sim \mu^m}\left[E_m\right] &= 1 - \mathbb{P}_{x_{1:m} \sim \mu^m}\left[\exists i < j \text{ such that } S_{ij} \leq \varepsilon m\right] \\
&\geq 1 - \sum_{i<j} \mathbb{P}_{x_{1:m} \sim \mu^m}\left[S_{ij} \leq \varepsilon m\right] \\
&\geq 1 - \sum_{i<j} \exp\left\{-2m\left(d_\mu(h_i, h_j) - \varepsilon\right)^2\right\} \\
&\geq 1 - \sum_{i<j} \exp\{-2m\,\delta^2\} \\
&\geq 1 - n^2 \exp\{-2m\,\delta^2\},
\end{aligned}$$

where the second inequality follows by Hoeffding's inequality. Moreover, under event $E_m$, we have that $\mathcal{M}(\varepsilon, \mathcal{H}, d_{\hat{\mu}_m}) \geq n$, where $d_{\hat{\mu}_m}$ is the empirical measure on $x_{1:m}$. Finally, note that

$$
\begin{aligned}
\sup_{m \in \mathbb{N}} \mathbb{E}_{x_{1:m} \sim \mu^m} \left[ \mathcal{M}(\varepsilon, \mathcal{H}, d_{\hat{\mu}_m}) \right] &\geq \sup_{m \in \mathbb{N}} \mathbb{E}_{x_{1:m} \sim \mu^m} \left[ \mathcal{M}(\varepsilon, \mathcal{H}, d_{\hat{\mu}_m}) | E_m \right] \mathbb{P}_{x_{1:m} \sim \mu^m} \left[ E_m \right] \\
&\geq n \sup_{m \in \mathbb{N}} \mathbb{P}_{x_{1:m} \sim \mu^m} \left[ E_m \right] \\
&\geq n \sup_{m \in \mathbb{N}} \left( 1 - n^2 \exp\{-2m \, \delta^2\} \right) \\
&= n.
\end{aligned}
$$

Since $n > c$ and $c = \sup_{m \in \mathbb{N}} \mathbb{E}_{x_{1:m} \sim \mu^m} \left[ \mathcal{M}(\varepsilon, \mathcal{H}, d_{\hat{\mu}_m}) \right]$, we arrive at a contradiction. $\qquad \square$

**Lemma B.3** (Covering Number of Symmetric Differences). *For any $\mathcal{H} \subseteq \mathcal{Y}^{\mathcal{X}}$, $\varepsilon > 0$, and sequence $x_1, ..., x_n \in \mathcal{X}^n$, we have that*

$$
\mathcal{N}(\varepsilon, \mathcal{H}\Delta\mathcal{H}, \hat{\mu}_n) \leq \left( \mathcal{N}\left(\frac{\varepsilon}{2}, \mathcal{H}, \hat{\mu}_n\right) \right)^2
$$

*where $\hat{\mu}_n$ is the empirical measure on $x_{1:n}$ and $\mathcal{H}\Delta\mathcal{H} := \left\{ x \mapsto \mathbb{1}\{h_1(x) \neq h_2(x)\} : h_1, h_2 \in \mathcal{H} \right\}$.*

*Proof.* Fix $\varepsilon > 0$. Let $\mathcal{H}'$ be an $\varepsilon$-cover for $\mathcal{H}$ with respect to $d_{\hat{\mu}_n}$. It suffices to show that $\mathcal{H}'\Delta\mathcal{H}'$ is an $2\varepsilon$-cover for $\mathcal{H}\Delta\mathcal{H}$ with respect to $d_{\hat{\mu}_n}$. To see this, pick a $g \in \mathcal{H}\Delta\mathcal{H}$. Then by definition, we can decompose $g(x) = \mathbb{1}\{h_1(x) \neq h_2(x)\}$ for some $h_1, h_2 \in \mathcal{H}$. Let $h_1', h_2' \in \mathcal{H}'$ be the elements that cover $h_1, h_2$ respectively. Then, observe that we have

$$
\frac{1}{n} \sum_{i=1}^n \mathbb{1}\{\mathbb{1}\{h_1'(x_i) \neq h_2'(x_i)\} \neq \mathbb{1}\{h_1(x_i) \neq h_2(x_i)\}\} \leq \frac{1}{n} \left( \sum_{i=1}^n \mathbb{1}\{h_1(x_i) \neq h_1'(x_i)\} + \sum_{i=1}^n \mathbb{1}\{h_2(x_i) \neq h_2'(x_i)\} \right)
$$

$$
\leq 2\varepsilon.
$$

Thus, $x \mapsto \mathbb{1}\{h_1'(x) \neq h_2'(x)\}$ is $2\varepsilon$-close to $x \mapsto \mathbb{1}\{h_1(x) \neq h_2(x_i)\}$. Since $x \mapsto \mathbb{1}\{h_1'(x) \neq h_2'(x)\} \in \mathcal{H}'\Delta\mathcal{H}'$ and $h_1, h_2 \in \mathcal{H}$ were chosen arbitrarily, it follows that $\mathcal{H}'\Delta\mathcal{H}'$ is a $2\varepsilon$-cover for $\mathcal{H}\Delta\mathcal{H}$ with respect to $d_{\hat{\mu}_n}$. Finally, note that $|\mathcal{H}'\Delta\mathcal{H}'| \leq |\mathcal{H}'|^2$. Since $\varepsilon > 0$ is arbitrary, this completes the proof. $\qquad \square$

## C Proof of Theorem 3.3

The proof here is similar to the proof of Theorem 3.1. Thus, we only provide a high-level sketch of the arguments here.

*Proof.* Fix $m \in \mathbb{N}$, and take $\mathcal{X} = \{1, 2, \ldots, m^2\}$. For every ordered sequence $(x_1, x_2, \ldots, x_m) \in \mathcal{X}^m$ such that $x_i \neq x_j$ for all $i \neq j$ and a bitstring $\theta \in \{0,1\}^m$, define a hypothesis

$$
h^\theta_{(x_1, \ldots, x_m)}(x) := \begin{cases} ((x_1, \ldots, x_m), \theta_{\leq t}) & \text{if } \exists t \in [m] \text{ such that } x = x_t \\ ((x_1, \ldots, x_m), \star) & \text{otherwise} \end{cases}
$$

Let $O_m := \{(x_1, x_2, \ldots, x_m) \in \mathcal{X}^m : x_i \neq x_j\}$ be the set of all ordered sequences of length $m$ with distinct elements. Then, we define our hypothesis class to be

$$
\mathcal{H} := \bigcup_{(x_1, \ldots, x_m) \in O_m} \bigcup_{\theta \in \{0,1\}^m} \left\{ h^\theta_{(x_1, \ldots, x_m)} \right\}.
$$

Here, the label space is $\mathcal{Y} := \cup_{h \in \mathcal{H}} \{\text{image}(h)\}$. Thus, the size of the label space is

$$
|\mathcal{Y}| = \binom{m^2}{m} m! \, 2^m = \frac{m^2!}{(m^2 - m)!} 2^m = m^2(m^2-1) \ldots (m^2-(m-1)) \, 2^m \leq (m^2)^m 2^m \leq m^{3m},
$$

when $m \geq 2$. This implies that

$$m \geq \frac{1}{3} \frac{\log |\mathcal{Y}|}{\log \log |\mathcal{Y}|}.$$

Note that the Proof of (i) in Theorem 3.1 can be used verbatim to show that $\mathcal{H}$ has a compression scheme of size 1. Thus, $\mathcal{H}$ is PAC learnable with error rate $O\left(\sqrt{\frac{\log n + \log(1/\delta)}{n}}\right)$.

Therefore, we now focus on establishing regret lowerbound for $\mathcal{H}$. Let $\mu = \text{Uniform}(\{1, 2, \ldots, m^2\})$. Let $m \leq T$. We now specify the stream $\{(x_t, y_t)\}_{t=1}^T$ to be observed by the learner. For the instances, we take $x_1, \ldots, x_m \sim \mu$ to be iid samples from $\mu$. Note that $x_1, \ldots, x_m$ are distinct with probability

$$\geq \left(1 - \frac{m}{m^2}\right)^m = \left(1 - \frac{1}{m}\right)^m \geq \frac{1}{4} \qquad \forall m \geq 2.$$

Moreover, as all the instances are drawn from the same distribution $\mu$, this adversary is $\sigma$-smooth for $\sigma = 1$. To specify $y_1, \ldots, y_m$, we first draw $\theta \sim \text{Uniform}(\{0, 1\}^m)$ and define $y_t = ((x_1, \ldots, x_m), \theta_{\leq t})$ for all $t \in [m]$. Conditioned on the fact that $x_1, \ldots, x_m$ are distinct, there will be a hypothesis class $h_\theta^* \in \mathcal{H}$ such that $h_\theta^*(x_t) = y_t$ for all $t \in [m]$. For $m \leq t \leq T$, sample $x_t \sim \mu$ and define $y_t = h_\theta^*(x_t)$. Note that for *any* $x_1, \ldots, x_m$, we have

$$\mathbb{E}_{\theta \sim \text{Uniform}(\{0,1\}^m)} \left[ \sum_{t=1}^T \mathbb{E}_{\mathcal{A}} \left[ \mathbb{1}\{\mathcal{A}(x_t) \neq y_t\} \right] \right] \geq \mathbb{E}_{\theta \sim \text{Uniform}(\{0,1\}^m)} \left[ \sum_{t=1}^m \mathbb{E}_{\mathcal{A}} \left[ \mathbb{1}\{\mathcal{A}(x_t) \neq y_t\} \right] \right] \geq \frac{m}{2}.$$

Here, we use the fact that bitstrings $\theta_t$ are sampled uniformly randomly, so no algorithm can do better than randomly guessing. Moreover, conditioned on the event that $x_1, \ldots, x_m$ are distinct, we have

$$\mathbb{E}_{\theta \sim \text{Uniform}(\{0,1\}^m)} \left[ \inf_{h \in \mathcal{H}} \sum_{t=1}^T \mathbb{1}\{h(x_t) \neq y_t\} \right] \leq \mathbb{E}_{\theta \sim \text{Uniform}(\{0,1\}^m)} \left[ \sum_{t=1}^T \mathbb{1}\{h_\theta^*(x_t) \neq y_t\} \right] = 0.$$

Since $x_1, \ldots, x_m$ are distinct with probability at least $1/8$, we obtain

$$\inf_{\mathcal{A}} \text{R}_{\mathcal{A}}^{\mu,\sigma}(T, \mathcal{H}) \geq \frac{m}{8} \geq \frac{1}{24} \frac{\log |\mathcal{Y}|}{\log \log |\mathcal{Y}|}.$$

$\square$

## D   Proof of Theorem 4.1

*Proof.* Let $\nu_1, \ldots, \nu_T \in \text{B}(\mu, \sigma)$ denote the sequence of $\sigma$-smooth distributions picked by the adversary. Fix a $\varepsilon > 0$. Then, by Lemma B.2, we have that $\mathcal{N}(\varepsilon, \mathcal{H}, d_\mu) \leq C_{\frac{\varepsilon}{2}, \sigma}(\mathcal{H}, \mu)$. Let $\mathcal{H}' \subset \mathcal{H}$ denote an $\varepsilon$-cover with respect to $d_\mu$ of size at most $C_{\frac{\varepsilon}{2}, \sigma}(\mathcal{H}, \mu)$. Let $\mathcal{A}$ denote the online learner that runs the Randomized Exponential Weights Algorithm (REWA) on the data stream $(x_1, y_1), \ldots, (x_T, y_T)$ using $\mathcal{H}'$ as its set of experts. By the guarantees of REWA,

$$\mathbb{E}_{\mathcal{A}} \left[ \sum_{t=1}^T \mathbb{1}\{\mathcal{A}(x_t) \neq y_t\} \right] \leq \inf_{h' \in \mathcal{H}'} \sum_{t=1}^T \mathbb{1}\{h'(x_t) \neq y_t\} + \sqrt{2T \log(|\mathcal{H}'|)}$$

$$\leq \inf_{h' \in \mathcal{H}'} \sum_{t=1}^T \mathbb{1}\{h'(x_t) \neq y_t\} + \sqrt{2T \log(C_{\frac{\varepsilon}{2}, \sigma}(\mathcal{H}, \mu))}$$

$$\leq \inf_{h \in \mathcal{H}} \sum_{t=1}^T \mathbb{1}\{h(x_t) \neq y_t\} + \sup_{h \in \mathcal{H}} \inf_{h' \in \mathcal{H}'} \sum_{t=1}^T \mathbb{1}\{h'(x_t) \neq h(x_t)\} + \sqrt{2T \log(C_{\frac{\varepsilon}{2}, \sigma}(\mathcal{H}, \mu))}$$

where the expectation is only taken with respect to the randomness of the MWA and the last inequality follows by the triangle inequality. Taking an outer expectation with respect to the process $x_{1:T} \sim \nu_{1:T}$,

we get that

$$\mathbb{E}\Big[\sum_{t=1}^{T}\mathbb{1}\{\mathcal{A}(x_t)\neq y_t\}\Big]$$

$$\leq \mathbb{E}\left[\inf_{h\in\mathcal{H}}\sum_{t=1}^{T}\mathbb{1}\{h(x_t)\neq y_t\}\right] + \mathbb{E}\left[\sup_{h\in\mathcal{H}}\inf_{h'\in\mathcal{H}'}\sum_{t=1}^{T}\mathbb{1}\{h'(x_t)\neq h(x_t)\}\right] + \sqrt{2T\log(C_{\frac{\varepsilon}{2},\sigma}(\mathcal{H},\mu))}.$$

It remains to bound $\mathbb{E}\left[\sup_{h\in\mathcal{H}}\inf_{h'\in\mathcal{H}'}\sum_{t=1}^{T}\mathbb{1}\{h'(x_t)\neq h(x_t)\}\right]$. To do so, consider the class $\mathcal{G}=\{x\mapsto\mathbb{1}\{h'(x)\neq h(x)\}:h\in\mathcal{H}\}$, where $h'\in\mathcal{H}'$ denotes the $\varepsilon$-cover with respect to $d_\mu$ of $h$, and note that

$$\mathbb{E}\left[\sup_{h\in\mathcal{H}}\inf_{h'\in\mathcal{H}'}\sum_{t=1}^{T}\mathbb{1}\{h'(x_t)\neq h(x_t)\}\right] \leq \mathbb{E}\left[\sup_{g\in\mathcal{G}}\sum_{t=1}^{T}g(x_t)\right].$$

By standard symmetrization arguments, we get that

$$\mathbb{E}_{x_{1:T}\sim\nu_{1:T}}\left[\sup_{g\in\mathcal{G}}\left(\sum_{t=1}^{T}g(x_t)-\mathbb{E}_{x'_{1:T}\sim\nu_{1:T}}\left[\sum_{t=1}^{T}g(x'_t)\right]\right)\right] \leq \mathbb{E}_{x_{1:T},x'_{1:T}\sim\nu_{1:T}}\left[\sup_{g\in\mathcal{G}}\sum_{t=1}^{T}\Big(g(x_t)-g(x'_t)\Big)\right]$$

$$=\mathbb{E}_{x_{1:T},x'_{1:T}\sim\nu_{1:T}}\left[\mathbb{E}_{\theta_{1:T}}\left[\sup_{g\in\mathcal{G}}\sum_{t=1}^{T}\theta_t\Big(g(x_t)-g(x'_t)\Big)\right]\right]$$

$$\leq 2\mathbb{E}_{x_{1:T}\sim\nu_{1:T}}\left[\mathbb{E}_{\theta_{1:T}}\left[\sup_{g\in\mathcal{G}}\sum_{t=1}^{T}\theta_t\,g(x_t)\right]\right]$$

$$\leq 2T\mathbb{E}_{x_{1:T}\sim\nu_{1:T}}\left[\hat{\mathfrak{R}}(\mathcal{G},x_{1:T})\right]$$

where $\theta_{1:T}$ are independent Rademacher random variables. Note that $\mathcal{G}\subseteq\mathcal{H}\Delta\mathcal{H}$ where $\mathcal{H}\Delta\mathcal{H}:=\{x\mapsto\mathbb{1}\{h_1(x)\neq h_2(x)\}:h_1,h_2\in\mathcal{H}\}$, and thus $\hat{\mathfrak{R}}(\mathcal{G},x_{1:T})\leq\hat{\mathfrak{R}}(\mathcal{H}\Delta\mathcal{H},x_{1:T})$. Using Lemma A.1 we can pointwise upperbound

$$\hat{\mathfrak{R}}(\mathcal{H}\Delta\mathcal{H},x_{1:T})\leq\varepsilon+\sqrt{\frac{2\log\mathcal{N}(\varepsilon,\mathcal{H}\Delta\mathcal{H},\rho_{\mu_T})}{T}}\leq\varepsilon+\sqrt{\frac{2\log\mathcal{N}(\varepsilon^2,\mathcal{H}\Delta\mathcal{H},d_{\mu_T})}{T}}\leq\varepsilon+2\sqrt{\frac{\log\mathcal{N}(\frac{\varepsilon^2}{2},\mathcal{H},d_{\mu_T})}{T}}.$$

The first inequality follows by taking $\rho_{\mu_T}(g_1,g_2)=\sqrt{\frac{1}{T}\sum_{t=1}^{T}\mathbb{1}\{g_1(x_t)\neq g_2(x_t)\}}$ for any two functions $g_1,g_2\in\mathcal{H}\Delta\mathcal{H}$. The second inequality follows from the fact that $\mathcal{N}(\varepsilon,\mathcal{H}\Delta\mathcal{H},\rho_{\mu_T})\leq\mathcal{N}(\varepsilon^2,\mathcal{H}\Delta\mathcal{H},d_{\mu_T})$. The last inequality follows after using Lemma B.3. Plugging in the upperbound on the Rademacher complexity, we get that

$$\mathbb{E}_{x_{1:T}\sim\nu_{1:T}}\left[\sup_{g\in\mathcal{G}}\sum_{t=1}^{T}g(x_t)\right]\leq\sup_{g\in\mathcal{G}}\mathbb{E}_{x'_{1:T}\sim\nu_{1:T}}\left[\sum_{t=1}^{T}g(x'_t)\right]+2\varepsilon\,T+4\mathbb{E}_{x_{1:T}\sim\nu_{1:T}}\left[\sqrt{T\log\mathcal{N}\left(\frac{\varepsilon^2}{2},\mathcal{H},d_{\mu_T}\right)}\right]$$

$$\leq\sup_{g\in\mathcal{G}}\sum_{t=1}^{T}\mathbb{E}_{x'_t\sim\nu_t}[g(x'_t)]+2\varepsilon\,T+4\sqrt{T\log\mathbb{E}_{x_{1:T}\sim\nu_{1:T}}\left[\mathcal{N}\left(\frac{\varepsilon^2}{2},\mathcal{H},d_{\mu_T}\right)\right]}$$

$$\leq\sup_{g\in\mathcal{G}}\sum_{t=1}^{T}\mathbb{E}_{x'_t\sim\mu}\left[\frac{g(x'_t)}{\sigma}\right]+2\varepsilon\,T+4\sqrt{T\log\mathbb{E}_{x_{1:T}\sim\nu_{1:T}}\left[\mathcal{N}\left(\frac{\varepsilon^2}{2},\mathcal{H},d_{\mu_T}\right)\right]}$$

$$\leq\frac{\varepsilon T}{\sigma}+2\varepsilon\,T+4\sqrt{T\log\mathbb{E}_{x_{1:T}\sim\nu_{1:T}}\left[\mathcal{N}\left(\frac{\varepsilon^2}{2},\mathcal{H},d_{\mu_T}\right)\right]},$$

where the third and fourth inequality follow by change of measure, $\sigma$-smoothness, and the definition $\mathcal{H}'$ respectively. By the definition of $C_{\varepsilon,\sigma}(\mathcal{H}, \mu)$, we get that

$$\mathbb{E}\left[\sup_{h\in\mathcal{H}}\inf_{h'\in\mathcal{H}'}\sum_{t=1}^{T}\mathbb{1}\{h'(x_t)\neq h(x_t)\}\right] \leq \frac{\varepsilon T}{\sigma} + 2\varepsilon\,T + 4\sqrt{T\,\log C_{\frac{\varepsilon^2}{2},\sigma}(\mathcal{H},\mu)},$$

implying that

$$\mathbb{E}\left[\sum_{t=1}^{T}\mathbb{1}\{\mathcal{A}(x_t)\neq y_t\}\right] \leq \mathbb{E}\left[\inf_{h\in\mathcal{H}}\sum_{t=1}^{T}\mathbb{1}\{h(x_t)\neq y_t\}\right] + \frac{\varepsilon T}{\sigma} + 2\varepsilon\,T + 4\sqrt{T\,\log C_{\frac{\varepsilon^2}{2},\sigma}(\mathcal{H},\mu)} + \sqrt{2T\,\log C_{\frac{\varepsilon}{2},\sigma}(\mathcal{H},\mu)}$$

$$\leq \mathbb{E}\left[\inf_{h\in\mathcal{H}}\sum_{t=1}^{T}\mathbb{1}\{h(x_t)\neq y_t\}\right] + \frac{3\varepsilon T}{\sigma} + 6\sqrt{T\,\log C_{\frac{\varepsilon^2}{2},\sigma}(\mathcal{H},\mu)}.$$

Since $\nu_1, \ldots, \nu_T \in \mathrm{B}(\mu, \sigma)$ and $\varepsilon > 0$ were chosen arbitrarily, we have that

$$\mathrm{R}_{\mathcal{A}}^{\mu,\sigma}(T,\mathcal{H}) \leq \inf_{\varepsilon>0}\left\{\frac{3\varepsilon T}{\sigma} + 6\sqrt{T\,\log C_{\frac{\varepsilon^2}{2},\sigma}(\mathcal{H},\mu)}\right\} \leq 6\inf_{\varepsilon>0}\left\{\frac{\varepsilon T}{\sigma} + \sqrt{T\,\log C_{\varepsilon^2,\sigma}(\mathcal{H},\mu)}\right\}.$$

$\square$

# E   Proof of Corollary 4.2

The following lemma from Haghtalab [2018], which uses the seminal packing lemma by Haussler [1995], will be useful.

**Lemma E.1** (Haussler [1995], Haghtalab [2018]). *For any $\mathcal{H} \subseteq \{0,1\}^{\mathcal{X}}$ and $\mu \in \Pi(\mathcal{X})$, we have that*

$$\mathcal{N}(\varepsilon, \mathcal{H}, d_\mu) \leq \left(\frac{41}{\varepsilon}\right)^{\mathrm{VC}(\mathcal{H})}.$$

Lemma E.1 together with Definition 6 implies that for any multiclass hypothesis class $\mathcal{H} \subseteq \mathcal{Y}^{\mathcal{X}}$, we have that

$$\sup_{\tilde{\mu}\in\Pi(\mathcal{X}\times\mathcal{Y})} \mathcal{N}(\varepsilon, \ell\circ\mathcal{H}, d_{\tilde{\mu}}) \leq \left(\frac{41}{\varepsilon}\right)^{\mathrm{G}(\mathcal{H})},$$

where $\ell\circ\mathcal{H} := \{(x,y)\mapsto\mathbb{1}\{h(x)\neq y\} : h\in\mathcal{H}\} \subseteq \{0,1\}^{(\mathcal{X}\times\mathcal{Y})}$ denotes the loss class of $\mathcal{H}$. We now begin the proof of Corollary 4.2.

*Proof.* (of Corollary 4.2) It suffices to show that

$$C_{\varepsilon,\sigma}(\mathcal{H},\mu) \leq \sup_{\tilde{\mu}\in\Pi(\mathcal{X}\times\mathcal{Y})} \mathcal{N}\left(\frac{2\varepsilon}{|\mathcal{Y}|}, \ell\circ\mathcal{H}, d_{\tilde{\mu}}\right),$$

for every $\varepsilon, \sigma > 0$. Fix $\varepsilon, \sigma > 0$. Recall that

$$C_{\varepsilon,\sigma}(\mathcal{H},\mu) := \sup_{n\in\mathbb{N}}\sup_{\nu_{1:n}\in\mathrm{B}(\mu,\sigma)} \mathop{\mathbb{E}}_{x_{1:n}\sim\nu_{1:n}}\left[\mathcal{N}(\varepsilon,\mathcal{H},d_{\hat{\mu}_n})\right],$$

where $\hat{\mu}_n$ denotes the empirical measure over $x_{1:n}$. We will actually show something stronger, that is

$$\sup_{n \in \mathbb{N}} \sup_{x_{1:n} \in \mathcal{X}^n} \mathcal{N}(\varepsilon, \mathcal{H}, d_{\hat{\mu}_n}) \leq \sup_{\tilde{\mu} \in \Pi(\mathcal{X} \times \mathcal{Y})} \mathcal{N}\left(\frac{2\varepsilon}{|\mathcal{Y}|}, \ell \circ \mathcal{H}, d_{\tilde{\mu}}\right). \tag{2}$$

Fix $n \in \mathbb{N}$ and let $c := \sup_{\tilde{\mu} \in \Pi(\mathcal{X} \times \mathcal{Y})} \mathcal{N}(\frac{2\varepsilon}{|\mathcal{Y}|}, \ell \circ \mathcal{H}, d_{\tilde{\mu}})$. To see why Inequality (2) is true, consider a sequence $x_{1:n} \in \mathcal{X}^n$ and let $\hat{\mu}_n$ denote the empirical measure on $x_{1:n}$. Let $\tilde{\mu}_n \in \Pi(\mathcal{X} \times \mathcal{Y})$ be the joint measure over $\mathcal{X} \times \mathcal{Y}$ defined procedurally by first sampling $x \in \hat{\mu}_n$ and then sampling the label $y \sim \text{Uniform}(\mathcal{Y})$. Since $\tilde{\mu}_n \in \Pi(\mathcal{X} \times \mathcal{Y})$, by definition of $c$, there exists a subset $\mathcal{H}' \subseteq \mathcal{H}$ of size at most $c$ such that $\ell \circ \mathcal{H}'$ is an $\frac{2\varepsilon}{|\mathcal{Y}|}$-cover for $\ell \circ \mathcal{H}$ with respect to $\tilde{\mu}_n$. It suffices to show that $\mathcal{H}'$ is an $\varepsilon$-cover for $\mathcal{H}$ with respect to $\hat{\mu}_n$. Fix a $h \in \mathcal{H}$ and let $h'$ be the element in $\mathcal{H}'$ such that $d_{\tilde{\mu}_n}(\ell \circ h, \ell \circ h') \leq \frac{2\varepsilon}{|\mathcal{Y}|}$. Then, by definition, we have that

$$
\begin{aligned}
\frac{2\varepsilon}{|\mathcal{Y}|} &\geq d_{\tilde{\mu}_n}(\ell \circ h, \ell \circ h') \\
&= \mathop{\mathbb{E}}_{(x,y) \sim \tilde{\mu}_n} \left[\mathbb{1}\{\ell \circ h(x,y) \neq \ell \circ h'(x,y)\}\right] \\
&= \frac{1}{n} \sum_{i=1}^{n} \frac{1}{|\mathcal{Y}|} \sum_{j=1}^{|\mathcal{Y}|} \mathbb{1}\{\mathbb{1}\{h(x_i) \neq j\} \neq \mathbb{1}\{h'(x_i) \neq j\}\} \\
&= \frac{2}{n|\mathcal{Y}|} \sum_{i=1}^{n} \mathbb{1}\{h(x_i) \neq h'(x_i)\} \\
&= \frac{2}{|\mathcal{Y}|} d_{\hat{\mu}_n}(h, h').
\end{aligned}
$$

Therefore $h'$ is $\varepsilon$-close to $h$ with respect to $d_{\hat{\mu}_n}$. Since $h$ was chosen arbitrarily, this is true for all $h \in \mathcal{H}$. Accordingly, $\mathcal{H}'$ is an $\varepsilon$-cover for $\mathcal{H}$ with respect to $d_{\hat{\mu}_n}$, implying that $\mathcal{N}(\varepsilon, \mathcal{H}, d_{\hat{\mu}_n}) \leq c$. Since $n$ and $x_{1:n}$ was also chosen arbitrarily, we have that $\sup_{n \in \mathbb{N}} \sup_{x_{1:n} \in \mathcal{X}^n} \mathcal{N}(\varepsilon, \mathcal{H}, d_{\hat{\mu}_n}) \leq c$, completing the proof.

Finally, upperbound $6 \inf_{\varepsilon > 0} \left\{\frac{\varepsilon T}{\sigma} + \sqrt{T \, G(\mathcal{H}) \log\left(\frac{41|\mathcal{Y}|}{\varepsilon^2}\right)}\right\} \leq 12\sqrt{T \, G(\mathcal{H}) \log\left(\frac{41 \, T \, |\mathcal{Y}|}{\sigma^2}\right)}$ follows by picking $\varepsilon = \frac{\sigma}{\sqrt{T}}$. $\qquad\square$

