# OpenReview forum: "Smoothed Online Classification can be Harder than Batch Classification"
_NeurIPS.cc/2024/Conference — NeurIPS 2024 poster_

### Official Review · Reviewer_SmVy · 2024-07-02

**Soundness:** 3
**Presentation:** 3
**Contribution:** 3
**Rating:** 7
**Confidence:** 1

**Summary:**

The paper studied online classification under smoothed adversaries. They constructed a hypothesis that is learnable under batch learning (PAC learning) but not learnable under smoothed online learning. They also showed that a sufficient condition that a hypothesis class learnable under the PAC learning is also learnable under the smoothed online learning.

**Strengths:**

- The result that smoothed online learning can be more difficult than PAC learning is interesting as it changes the conventional ideas in the community. As smoothed (non-adversarial) online learning gets more important as large models are emerging, the result may also contribute the theoretical insights for such models.

**Weaknesses:**

- They didn’t show a necessary and sufficient condition that a hypothesis class learnable under the PAC learning is also learnable under the smoothed online learning.

**Questions:**

N/A

---

> ### Author Rebuttal · Authors · 2024-08-05
>
> We thank the reviewer for their comments.
>
> We are unsure what the reviewer meant by "They didn’t show a necessary and sufficient condition that a hypothesis class learnable under the PAC learning is also learnable under the smoothed online learning." We would like to point out that the main result of our work shows that PAC learnability is provably not sufficient for smoothed online classification for unbounded label spaces. However, PAC learnability is certainly necessary for smoothed online classification for unbounded label spaces.

---

> > ### Comment · Reviewer_SmVy · 2024-08-13
> >
> > Thank you for the correction. I misunderstood the definition of PAC learnability.
> > I will keep my current score, but other reviewers probably have better understanding of the paper.

---

### Official Review · Reviewer_ercW · 2024-07-07

**Soundness:** 3
**Presentation:** 3
**Contribution:** 2
**Rating:** 4
**Confidence:** 3

**Summary:**

This paper studies online learning under the smoothed analysis framework. Under the smoothed analysis framework, the example that is presented to the learner in each round is not chosen adversarially but instead is drawn from some distribution that is close to a known based distribution. Previous works have shown that for binary classification or regression tasks if a hypothesis class can be learned under the PAC learning model, then it can also be learned under the smoothed online learning model. In this work, the authors consider multi-class classification against a non-adaptive adversary. They show that if the label class $\mathcal{Y}$ is unbounded, then one can construct a hypothesis class $H$, which has a finite sample compression scheme (and is thus PAC learnable) but is not smoothed online learnable. On the other hand, the paper extends previous results on binary classification and proposes a sufficient condition that ensures the PAC
 learnability of a hypothesis class is sufficient for its smoothed online learnability.

**Strengths:**

The paper studies a well-defined learning problem that combines multiclass classification with smoothed online learning and gives several results on the learnability of the problem. These results enhance our understanding of the smoothed online learning problem. The hardness result for the case where the label class $\mathcal{Y}$ is unbounded is interesting and non-trivial.

**Weaknesses:**

My main concern is about the significance of the work.

The learning model studied in this paper competes with a non-adaptive adversary.
Though the paper generalizes the condition for the learnability of the problem from binary classification to multi-class classification, the high-level idea of the proof seems to be a straightforward extension of the analysis used in Haghtalab's thesis (though a more complicated analysis is needed due to the more complicated setting). Furthermore, for binary classification/regression, as mentioned in the introduction of the paper, there are results that can compete with an even stronger adaptive adversary. Given these, I am not quite convinced about the importance of the sufficient condition on the learnability proposed by the paper.

On the other hand, the hard instance is constructed based on a not very extensively studied learning model, multiclassification with infinite label sets. Though the author points out several very recent papers that study this setting, it is still not very clear to me the motivation for studying the setting and how such a hardness result would inspire new theories or algorithms for related problems.

**Questions:**

I would like the authors to comment on the weakness pointed out above.

---

> ### Author Rebuttal · Authors · 2024-08-03
>
> We thank the reviewer for noting that our results "enhance our understanding of the smoothed online learning problem." We address their concerns below.
>
> - Our sufficiency condition strictly improves the sufficiency condition presented in Haghtalab's thesis, as noted in lines 312-316. Specifically, there are classes where the VC dimension is infinite and the upper bounds in Haghtalab's thesis are therefore vacuous, but our sufficiency condition still provides a meaningful upper bound. We believe that the significance of our sufficiency condition is more conceptual than technical.  Our upper bound highlights that distribution-dependent complexity measures quantify the rates more precisely than distribution-independent complexity measures. This is an important point worth highlighting because any complexity measure that characterizes smoothed online learnability has to be a joint property of both $\mathcal{H}$ and $\mu$ (as noted in the Discussion section). Lastly, we only considered non-adaptive adversaries because the primary focus of our work is the hardness result. However, our upper bounds can be extended to adaptive adversaries using the coupling argument from [1].
>
> -  We note that Theorem 3.3 shows that, in terms of regret bound, the separation between PAC learnability and smoothed online learnability holds even for finite label spaces as long as the size of the label space is $\geq 2^{T \log(T)}$. More precisely, for any time horizon $T$, there exists a class $\mathcal{H}_T \subseteq \mathcal{Y}^{\mathcal{X}}$ with $|\mathcal{Y}|\geq 2^{T \log{T}}$ for which its PAC error bound is $O\left(\sqrt{\frac{\log{n}}{n}} \right)$, but its regret is $\Omega(T)$ (or average regret is $\Omega(1)$). That is, $\mathcal{H}_T$ has a vanishing PAC upper bound (independent of $T$) but a constant non-vanishing average regret lower bound. The infinite label space is required only to show a separation of \emph{learnability}. In particular, showing that the class is non-learnable according to Definition 1 requires establishing $\Omega(1)$ average regret for every $T$ even as $T \to \infty$.
>
>
> - Finally, we disagree with the reviewer that multiclass classification with infinite label sets is not ``a not very extensively studied learning model." This setting has been studied in several seminal works in learning theory in the last 40 years beginning with [2,3] and more recently in [4,5,6] to name a few. Studying infinite label spaces is important for understanding when one can establish learning guarantees independent of the label size. This is quite a practical question as many modern machine learning paradigms have massive label space, such as in face recognition and protein structure prediction, where the dependence of label size in learning bounds would be undesirable.
>
>
> [1] Haghtalab, Nika, Tim Roughgarden, and Abhishek Shetty. ``Smoothed analysis with adaptive adversaries." Journal of the ACM 71.3 (2024): 1-34.
>
>
> [2] Balas K. Natarajan. Some results on learning. 1988.
>
> [3] B. K. Natarajan. On learning sets and functions. Machine Learning, 4:67–97, 1989.
>
> [4] A. Daniely, S. Sabato, S. Ben-David, and S. Shalev-Shwartz. Multiclass learnability and the ERM principle. In Proceedings of the 24th Conference on Learning Theory, 2011.
>
> [5] A. Daniely and S. Shalev-Shwartz. Optimal learners for multiclass problems. In Proceedings of the 27th Conference on Learning Theory, 2014.
>
> [6] N. Brukhim, D. Carmon, I. Dinur, S. Moran, and A. Yehudayoff. A characterization of multiclass learnability. In Proceedings of the 63rd Annual IEEE Symposium on Foundations of Computer Science, 2022.

---

> > ### Comment · Reviewer_ercW · 2024-08-11
> >
> > I would like to thank the authors for making comments on my review and providing related references. However, I am still not very convinced about the significance of the work and I am not sure how much interest the setting smoothed online learning with infinite labels could attract from the NeurIPS community. I think more work should be done and included to convince readers that this setting is not a simple combination of smoothed online learning and multiclass classification with infinite labels. For this reason, I will keep my current rate.

---

> > > ### Author Response · Authors · 2024-08-12
> > >
> > > We thank the reviewer for their response and would like to respond to some of their concerns.
> > >
> > > > "Still not very convinced about the significance of the work"
> > >
> > > Both smoothed learning and multiclass classification with infinite labels are recent but well-established topics. Smoothed analysis originated in the analysis of one of best known problem in CS, viz. linear programming. In recent years, smoothed analysis has been extended to learning problems. Multiclass classification  is one of the most fundamental ML problems. The significant of infinite labels is both theoretical and practical. Theoretically, it was the infinite labels setting that led to the recent complete characterization of multiclass learnability. From a practical point of view, infinite labels is a means to study what happens with extremely large label spaces. This is relevant to work in NLP (large vocabularies) and in extreme multiclass classification (e.g., recommender systems).
> > >
> > > > "how much interest the setting … could attract from the NeurIPS community”
> > >
> > > We note that there have been several NeurIPS papers regarding smoothed online learning, even dating back to 2011 [1-4]. There is also a long history of NeurIPS papers studying classification with extremely large label spaces. This line of work is known as ``Extreme Classification" [5-11]. Thus, we think studying the intersection of these two settings is natural and of interest to the NeurIPS community.
> > >
> > > > "this setting is not a simple combination of smoothed online learning and multiclass classification with infinite labels”
> > >
> > > We respectfully disagree with this criticism. Yes, the setting is simple — it combined smoothed online learning with multiclass classification with infinite labels. But the analysis is far from simple and our conclusions are far from obvious. We think that a simple setting with non-obvious analysis and conclusions should be of interest to NeurIPS. We think that the simplicity of the setting should not be a drawback of our contribution.
> > >
> > > [1] Alexander Rakhlin, Karthik Sridharan, and Ambuj Tewari. Online learning: Stochastic, constrained,
> > > and smoothed adversaries. Advances in neural information processing systems, 24, 2011.
> > >
> > > [2] Nika Haghtalab. Foundation of Machine Learning, by the People, for the People. PhD thesis, Carnegie
> > > Mellon University, 2018.
> > >
> > > [3] Nika Haghtalab, Tim Roughgarden, and Abhishek Shetty. Smoothed analysis of online and differentially private learning. Advances in Neural Information Processing Systems, 33:9203–9215,379
> > > 2020.
> > >
> > > [4] Adam Block, Yuval Dagan, Noah Golowich, and Alexander Rakhlin. Smoothed online learning is as
> > > easy as statistical learning. In Conference on Learning Theory, pages 1716–1786. PMLR, 2022
> > >
> > > [5] K. Bhatia, H. Jain, P. Kar, M. Varma, and P. Jain, Sparse Local Embeddings for Extreme Multi-label Classification, in NeurIPS 2015.
> > >
> > > [6] D. Hsu, S. Kakade, J. Langford, and T. Zhang, Multi-Label Prediction via Compressed Sensing, in NeurIPS 2009.
> > >
> > > [7] Y. Chen, and H. Lin, Feature-aware Label Space Dimension Reduction for Multi-label Classification, in NeurIPS, 2012.
> > >
> > > [8] M. Cisse, N. Usunier, T. Artieres, and P. Gallinari, Robust Bloom Filters for Large Multilabel Classification Tasks , in NIPS, 2013.
> > >
> > > [9] I. Evron, E. Moroshko and K. Crammer, Efficient Loss-Based Decoding on Graphs for Extreme Classification in NeurIPS, 2018.
> > >
> > > [10] R. You, S. Dai, Z. Zhang, H. Mamitsuka, and S. Zhu, AttentionXML: Extreme Multi-Label Text Classification with Multi-Label Attention Based Recurrent Neural Network, in NeurIPS 2019.
> > >
> > > [11] S. Kharbanda, A. Banerjee, R. Schultheis and R. Babbar, CascadeXML : Rethinking Transformers for End-to-end Multi-resolution Training in Extreme Multi-Label Classification, in NeurIPS 2022.

---

### Official Review · Reviewer_jMfw · 2024-07-09

**Soundness:** 3
**Presentation:** 3
**Contribution:** 2
**Rating:** 6
**Confidence:** 4

**Summary:**

This paper studies the problem of distinguishing batch learning from smoothed online learning when the label set size is unbounded. It shows that there exists a class that can be PAC-learned but does not admit sublinear regret, even with features generated i.i.d. The paper then provides a sufficient condition based on an empirical covering number that guarantees sublinear regret for smoothed adversarial learning.

**Strengths:**

The contribution of this paper is primarily conceptual. It demonstrates that while the learnability of hypotheses with bounded label sets is fairly well understood, there can be mysterious behavior in the unbounded label case that requires further attention. From a technical standpoint, this paper constructs several instances that demonstrate separations which may be of independent interest. Overall, this is an interesting paper in a niche area that is suitable for the NeurIPS community.

**Weaknesses:**

While I do like the philosophical message delivered by this paper, from a purely technical standpoint, it feels somewhat half-baked.

I outline some specific comments below:

1. While the title indicates "Smoothed Online Classification can be Harder than Batch Classification," the real separation here is actually between batch learning and *adversarial* (label) online learning. The example constructed employs no property of the smoothed adversarial setting.

2. While the paper provides a sufficient condition for smoothed online learning, it does not demonstrate how strong this condition is. Is it necessary as well? Theorem 4.3 (ii) seems to indicate this is not the case.

3. There are many relevant problems that should have been studied in this paper but are unfortunately omitted or left to future work. See the Questions section below.

**Questions:**

1. It seems all your hard instances boil down to dealing with *adversarial* labels. Can the authors comment on what happens for *realizable* labels? Does the separation still hold? (Note that in this case, you will have to use properties of smooth adversaries, as i.i.d. features are trivial.)

2. It seems the failure in Theorem 4.3 (ii) is due to the non-sequential nature of your empirical cover. Can a similar notion, such as the stochastic sequential cover from Wu et al., [2023] (perhaps using the approximate variant from Hanneke et al., [2023]), be sufficient to characterize the learnability?

3. Does the learnability of smoothed adversarial in the realizable case imply learnability in the adversarial label case, perhaps using a similar sequential cover construction from Hanneke et al., [2023]?

---

> ### Author Rebuttal · Authors · 2024-08-03
>
> We thank the reviewers for their comments and address their concerns below.
>
> **Weaknesses**
> 1.  In online classification, smoothness is only an assumption on the instances $x_1, ..., x_T$, and the labels can still be adversarial. This is the standard smooth model considered in
> [1,2,3,4]. If by 'adversarial labels' the reviewer means noisy labels, then we would like to clarify that our hardness result is actually derived in the realizable setting. This is established in the math display between lines 280 and 281, where we show the existence of a hypothesis that perfectly labels the data and achieves $0$ cumulative loss.
>
> 2. The reviewer is correct in that the sufficient condition is not necessary. We mention this fact in lines 97-99 and formalize it in Theorem 4.3. As noted in the Discussion section, the smoothed model allows some pathological edge cases that make characterizing learnability difficult.
>
> **Questions**
>
> 1. We would like to clarify that our hardness result is in the realizable setting. This is established in the math display between lines 280 and 281, where we show the existence of a hypothesis that perfectly labels the data and achieves $0$ cumulative loss.
>
> 2. The notion of stochastic sequential cover provided in  Definition 1 of (Wu et al. 2023) does not characterize smoothed learnability. In fact, this can be established using the same examples used in the proof of Theorem 4.3. To see why it is not sufficient, consider  $\mathcal{X}=[0,1]$ and $\mathcal{H}= \lbrace{ x \mapsto 1[x \in S]\, : S \subset \mathcal{X} \text{ and } |S|<\infty\rbrace}$. Let $\mathcal{P}$ denote set of all distributions on $\mathcal{X}^{T}$ such that for any $\nu \in \mathcal{P}$, its marginal at each $t$ is $\sigma$-smooth with respect to $\mu=\text{Uniform}(\mathcal{X})$ for some fixed $\sigma>0$. It is easy to see that the size of the stochastic sequential cover of $\mathcal{H}$ with respect to $\mathcal{P}$ under $0$-$1$ metric is $1$ (see proof of Theorem 4.3 (i) for details). However, as shown in Theorem 4.3 (i), this class is not learnable under the smoothed model. On the other hand, the class $\mathcal{H} = \lbrace{x \mapsto a: a \in \mathbb{N}\rbrace}$ shows that the stochastic sequential cover is not necessary, as its size of sequential cover for any distribution family $\mathcal{P}$ is $\infty$.
>
> 3. Realizable and agnostic learnability are equivalent for smoothed online classification even when the label space is unbounded. We have some ongoing work that establishes this result using an adaptation of Theorem 11 in [6] for infinite label spaces.
>
> [1] Alexander Rakhlin, Karthik Sridharan, and Ambuj Tewari. Online learning: Stochastic, constrained, and smoothed adversaries. Advances in neural information processing systems, 24, 2011.
>
> [2] Nika Haghtalab. Foundation of Machine Learning, by the People, for the People. PhD thesis, Carnegie Mellon University, 2018.
>
> [3] Nika Haghtalab, Tim Roughgarden, and Abhishek Shetty. Smoothed analysis of online and differentially private learning. Advances in Neural Information Processing Systems, 33:9203–9215,379
> 2020.
>
> [4] Adam Block, Yuval Dagan, Noah Golowich, and Alexander Rakhlin. Smoothed online learning is as
> easy as statistical learning. In Conference on Learning Theory, pages 1716–1786. PMLR, 2022.
>
> [5] Wu C, Heidari M, Grama A, Szpankowski W. Expected Worst Case Regret via Stochastic Sequential Covering. Transactions on Machine Learning Research, 2023.
>
> [6] Raman, Vinod, Unique Subedi, and Ambuj Tewari. "A Characterization of Multioutput Learnability." arXiv preprint arXiv:2301.02729 (2023).

---

> > ### Comment · Reviewer_jMfw · 2024-08-07
> >
> > I thank the authors for the clarification on the realizability. However, I still feel this work is not quite "complete"; perhaps a more theoretically oriented conference (such as COLT/ALT) will better appreciate it. Anyway, I have adjusted my rating to 6, but I will not fight for acceptance.

---

> > > ### Author Response · Authors · 2024-08-12
> > >
> > > We thank the reviewer for their response.
> > >
> > > We would like to point out that the smoothed online learning model was first studied by [1] in NeurIPS 2011 with the goal of bridging theory and practice. In addition, there is also a long history of NeurIPS papers studying classification with extremely large label spaces. This line of work is known as ``Extreme Classification" [2-7]. So, we think that our work is still of interest to the NeurIPS community.
> > >
> > > [1] Alexander Rakhlin, Karthik Sridharan, and Ambuj Tewari. Online learning: Stochastic, constrained, and smoothed adversaries. Advances in neural information processing systems, 24, 2011.
> > >
> > > [2] K. Bhatia, H. Jain, P. Kar, M. Varma, and P. Jain, Sparse Local Embeddings for Extreme Multi-label Classification, in NeurIPS 2015.
> > >
> > > [3] D. Hsu, S. Kakade, J. Langford, and T. Zhang, Multi-Label Prediction via Compressed Sensing, in NeurIPS 2009.
> > >
> > > [4] Y. Chen, and H. Lin, Feature-aware Label Space Dimension Reduction for Multi-label Classification, in NeurIPS, 2012.
> > >
> > > [5] M. Cisse, N. Usunier, T. Artieres, and P. Gallinari, Robust Bloom Filters for Large Multilabel Classification Tasks , in NIPS, 2013.
> > >
> > > [6] I. Evron, E. Moroshko and K. Crammer, Efficient Loss-Based Decoding on Graphs for Extreme Classification in NeurIPS, 2018.
> > >
> > > [7] R. You, S. Dai, Z. Zhang, H. Mamitsuka, and S. Zhu, AttentionXML: Extreme Multi-Label Text Classification with Multi-Label Attention Based Recurrent Neural Network, in NeurIPS 2019.
> > >
> > > [11] S. Kharbanda, A. Banerjee, R. Schultheis and R. Babbar, CascadeXML : Rethinking Transformers for End-to-end Multi-resolution Training in Extreme Multi-Label Classification, in NeurIPS 2022.

---

### Official Review · Reviewer_7N9e · 2024-07-12

**Soundness:** 3
**Presentation:** 3
**Contribution:** 3
**Rating:** 6
**Confidence:** 3

**Summary:**

They consider the problem of smoothed online classification under oblivious adversaries. From earlier work it is known that this problem is as easy as batch classification when the label space is bounded. However, when the label space is unbounded they provide a lower bound and show that this problem can be harder than batch classification in PAC model.

Furthermore, they provide a sufficient condition for smoothed online learnability. Their conditions are in terms of covering/packing numbers of the hypothesis class using a distance metric that depends on the base measure $\mu$. However, their sufficient condition is not a necessary condition for the smoothed learnability. They leave it as an open question to find a condition that is both necessary and sufficient for smoothed online learnability.

**Strengths:**

I did not go through all the proofs, but overall this paper is well-written and seems to be sound.

**Weaknesses:**

same as above.

**Questions:**

Minor comments:
\Sigma is not defined in line 118.

**Limitations:**

same as above.

---

> ### Author Rebuttal · Authors · 2024-08-03
>
> We thank the reviewer for finding our paper well-written. We will make sure to define $\Sigma$ in the camera-ready version.

---

> > ### Comment · Reviewer_7N9e · 2024-08-12
> >
> > I went through the other reviews and responses, and I agree that a more theoretically oriented venue like ALT/COLT might be a better fit. I keep my current score.

---

> > > ### Author Response · Authors · 2024-08-12
> > >
> > > We thank the reviewer for their response.
> > >
> > > We would like to point out that the smoothed online learning model was first studied by [1] in NeurIPS 2011 with the goal of bridging theory and practice. In addition, there is also a long history of NeurIPS papers studying classification with extremely large label spaces. This line of work is known as ``Extreme Classification" [2-7]. So, we think that our work is still of interest to the NeurIPS community.
> > >
> > > [1] Alexander Rakhlin, Karthik Sridharan, and Ambuj Tewari. Online learning: Stochastic, constrained, and smoothed adversaries. Advances in neural information processing systems, 24, 2011.
> > >
> > > [2] K. Bhatia, H. Jain, P. Kar, M. Varma, and P. Jain, Sparse Local Embeddings for Extreme Multi-label Classification, in NeurIPS 2015.
> > >
> > > [3] D. Hsu, S. Kakade, J. Langford, and T. Zhang, Multi-Label Prediction via Compressed Sensing, in NeurIPS 2009.
> > >
> > > [4] Y. Chen, and H. Lin, Feature-aware Label Space Dimension Reduction for Multi-label Classification, in NeurIPS, 2012.
> > >
> > > [5] M. Cisse, N. Usunier, T. Artieres, and P. Gallinari, Robust Bloom Filters for Large Multilabel Classification Tasks , in NIPS, 2013.
> > >
> > > [6] I. Evron, E. Moroshko and K. Crammer, Efficient Loss-Based Decoding on Graphs for Extreme Classification in NeurIPS, 2018.
> > >
> > > [7] R. You, S. Dai, Z. Zhang, H. Mamitsuka, and S. Zhu, AttentionXML: Extreme Multi-Label Text Classification with Multi-Label Attention Based Recurrent Neural Network, in NeurIPS 2019.
> > >
> > > [11] S. Kharbanda, A. Banerjee, R. Schultheis and R. Babbar, CascadeXML : Rethinking Transformers for End-to-end Multi-resolution Training in Extreme Multi-Label Classification, in NeurIPS 2022.

---

### Decision · Program_Chairs · 2024-09-25

**Decision:**

Accept (poster)

**Comment:**

During the discussion, the reviewers agreed that the paper is solid overall. However, concerns were raised about whether the theoretical framework of multiclass online classification with an infinite label space is properly justified within the paper. The reviewers emphasized the importance of the authors addressing this issue honestly and thoroughly in the final version to ensure the soundness of the work.